# FROM MOLECULES TO MATERIALS: PRE-TRAINING LARGE GENERALIZABLE MODELS FOR ATOMIC PROPERTY PREDICTION

**Nima Shoghi**[*][1]   **Adeesh Kolluru**[2]   **John R. Kitchin**[2]
**Zachary W. Ulissi**[1]   **C. Lawrence Zitnick**[1]   **Brandon M. Wood**[1]

[1]Fundamental AI Research (FAIR) at Meta
[2]Carnegie Mellon University
[*]Work done while at FAIR
Correspondence to: `ns@nima.sh, bmwood@meta.com`

## ABSTRACT

Foundation models have been transformational in machine learning fields such as natural language processing and computer vision. Similar success in atomic property prediction has been limited due to the challenges of training effective models across multiple chemical domains. To address this, we introduce Joint Multi-domain Pre-training (JMP), a supervised pre-training strategy that simultaneously trains on multiple datasets from different chemical domains, treating each dataset as a unique pre-training task within a multi-task framework. Our combined training dataset consists of ~120M systems from OC20, OC22, ANI-1x, and Transition-1x. We evaluate performance and generalization by fine-tuning over a diverse set of downstream tasks and datasets including: QM9, rMD17, MatBench, QMOF, SPICE, and MD22. JMP demonstrates an average improvement of 59% over training from scratch and matches or sets state-of-the-art on 34 out of 40 tasks. Our work highlights the potential of pre-training strategies that utilize diverse data to advance property prediction across chemical domains, especially for low-data tasks.

## 1   INTRODUCTION

Computing atomic properties accurately and efficiently for a vast array of molecules and materials is crucial for a range of applications, from drug discovery (Chan et al., 2019; Deng et al., 2022) to catalyst design (Zitnick et al., 2020). Currently, the quantum chemistry method Density Functional Theory (DFT) is commonly employed for atomic property calculations. Unfortunately, DFT's use is limited by its significant computational expense, which can range from hours to days for certain calculations. Machine learning (ML) potentials, which approximate or augment DFT, are capable of reducing the computational cost by orders of magnitude (Behler, 2016). In recent years, much progress has been made towards this goal (Kolluru et al., 2022b), fueled in part by the release of large and diverse DFT-generated datasets for training ML models. While these datasets are incredibly useful, they are also extremely expensive to generate, e.g., ~400 million CPU hours for the Open Catalyst 2020 dataset (OC20) (Chanussot et al., 2021). As a consequence, it is impractical to create a large dataset for every specific chemistry problem of interest. Similarly, it is non-ideal to train a model from scratch for all use cases, which is common practice currently.

Foundation models (FMs) — large pre-trained models that can be fine-tuned for various tasks — have achieved remarkable success in domains such as natural language processing (NLP) and computer vision (CV), especially when fine-tuned on low-resource downstream tasks. Several key factors have enabled this effectiveness: (1) the availability of massive datasets, (2) the development of widely adopted pre-training strategies, and (3) the establishment of diverse benchmarks to rigorously assess the performance of these fine-tuned models. Despite the availability of large DFT-labeled datasets (e.g., OC20) and the existence of a wide and diverse range of downstream tasks (e.g., QM9 (Ruddigkeit et al., 2012b), MatBench (Dunn et al., 2020)), the adoption of pre-training in ML

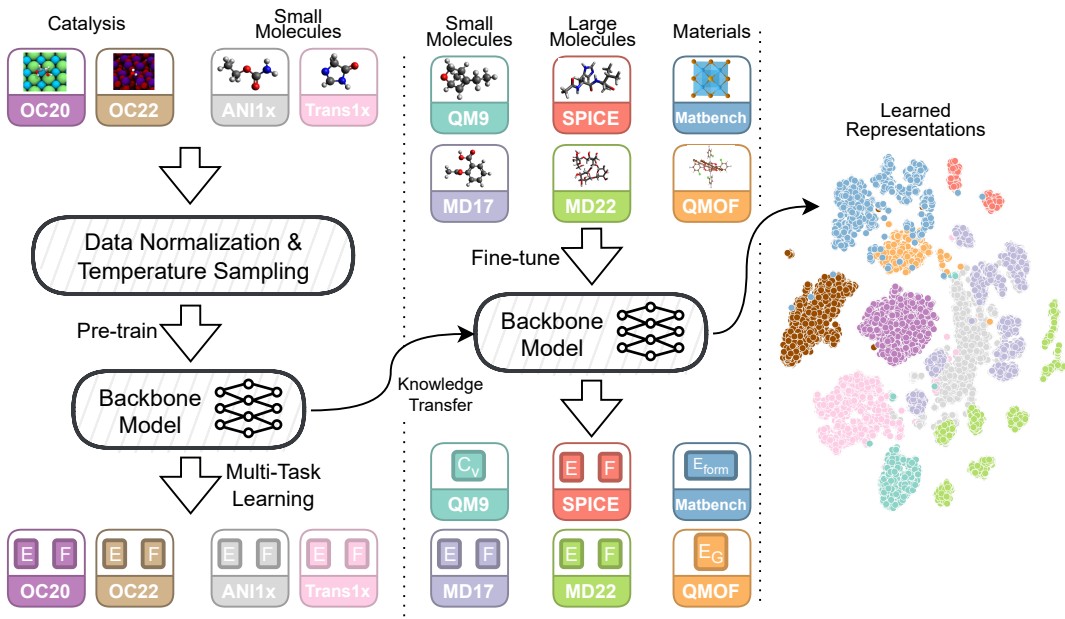

Figure 1: An overview of the Joint Multi-domain Pre-training (JMP) method. **Left**: JMP's pre-training setup, where a single model is simultaneously trained on set of diverse pre-training datasets using multi-task learning. **Center**: JMP's fine-tuning process, where the pre-trained JMP backbone is equipped with new prediction heads and trained on downstream tasks. **Right**: t-SNE visualizations of JMP's node-level ($\tilde{h}$) embeddings for randomly selected structures from all datasets.

for atomic property prediction has been noticeably less prevalent. This under-utilization becomes evident when noting that most of the state-of-the-art (SOTA) results on downstream tasks come from models trained from scratch. More specifically, prior to our work, all previous SOTA results on the rMD17, MD22, SPICE, and MatBench datasets come from models trained form scratch. For QM9, models trained from scratch hold SOTA status for 7 of the 12 targets. In total, out of the 40 total tasks explored in this work's evaluation benchmark, models trained from scratch hold the previous SOTA on 34 tasks.

At its core, the challenge of pre-training for atomic property prediction lies in the complexity and diversity of the underlying chemical space. Target applications vary from drug design to catalysis, the data ranges from small molecules with only 4 atoms to periodic crystals with hundreds, and even the properties of interest for each application vary from various energies to forces to phonon peaks. Furthermore, the nature of atomic properties imposes a unique set of challenges. Unlike in NLP or CV, where the data is often discrete and finite, atomic properties are continuous and can span several orders of magnitude. This requires models to be robust to outliers and capable of predicting highly variable outputs. Further, existing pre-training strategies (e.g., Zaidi et al. (2022); Zhou et al. (2023b)) are designed with equilibrium systems in mind and are not directly applicable to non-equilibrium systems, which are common in DFT datasets (e.g., over 99.7% of OC20's training data comprises of non-equilibrium structures). These challenges motivate the need for a flexible and generalizable pre-training strategy that can be adapted to different applications and datasets.

In this work, we introduce Joint Multi-domain Pre-training (JMP), a supervised pre-training strategy tailored to the challenges and opportunities of machine learning for atomic modeling. JMP concurrently trains over 120 million diverse equilibrium and non-equilibrium atomic structures by framing each chemical domain as a separate pre-training task in a multi-task framework. This large-scale pre-training enables learning generalizable representations of atomic interactions. The contributions of our work are summarized as follows: **First**, we introduce the JMP method, shown in Figure 1, and demonstrate its powerful generalization ability by evaluating its fine-tuning performance across a diverse benchmark suite spanning small molecules, large molecules, and materials. Our results show that JMP consistently outperforms training from scratch and sets or matches the **state-of-the-art on 34 out of the 40 fine-tuning benchmarks**. **Second**, we show that JMP enables efficient scaling to larger models that would normally overfit if trained from scratch on small datasets. Pre-training acts as a strong regularizer, allowing us to train a 235M parameter model that sets new state-of-the-art performance on multiple low-data benchmarks. **Finally**, we conduct a detailed analysis of JMP's

computational requirements. While expensive upfront, we show JMP's pre-training cost is recovered by enabling over 12x faster fine-tuning compared to training from scratch. By pre-training large models on diverse chemical data, we believe JMP represents an important step towards the goal of a universal ML potential, and that the continued growth of available data and compute power will only improve JMP's ability to learn transferable atomic representations.

## 2 RELATED WORK

**Machine learning potentials:** There has been significant progress in developing ML models for atomic property prediction. Initial approaches focused on descriptor-based methods, where these descriptors were hand-fitted physically meaningful analytical functions (González, 2011; Sundius, 2002; Dinur and Hagler, 1991). These functions were incorporated into gaussian process models (Chmiela et al., 2017) or neural networks (Behler and Parrinello, 2007). Recent advances in graph neural networks (GNNs) have shown to be a promising approach for these tasks, surpassing descriptor-based methods (Gasteiger et al., 2020; Schütt et al., 2017; Batzner et al., 2021; Batatia et al., 2022) on multiple benchmarks across the atomic domains of small molecules, catalysts, and bulk materials. While much progress has been made, it remains difficult for a single model to perform well across all chemical domains.

**Pretraining and transfer learning on 3D atomic systems:** The concept of transfer learning, where representations are learned on one dataset and transferred to another, has been successfully applied to a number of atomic modeling tasks (Kolluru et al., 2022a; Cai et al., 2020; Tsubaki and Mizoguchi, 2021; Smith et al., 2018). However, most of the focus in this area has been on transferring representations within the same chemical domain with a limited amount of pre-training data (Smith et al., 2019; Yamada et al., 2019; Pesciullesi et al., 2020). There are beginning to be more dedicated works on pre-training (Zhu et al., 2022; Liu et al., 2021; Jiao et al., 2022; Zhou et al., 2023b), but most do not explore generalization across multiple chemical domains. Many of these works focus on self-supervised pre-training on molecular graphs and/or 3D atomic structures. Recent self-supervised methods have focused on denoising methods (Song et al., 2020) applied to equilibrium structures — i.e. the per atom forces are close to zero (Zaidi et al., 2022; Feng et al., 2023b; Liu et al., 2022). The original formulation of denoising equilibrium structures is applicable to less than 1% of our training data because most of the atomic properties data is non-equilibrium. This is an active area of research and since the beginning of our present work, alternative formulations that could apply to non-equilibrium data have started to emerge (Feng et al., 2023a; Zheng et al., 2023).

## 3 DATASETS

We separate the atomic space into four domains for the purposes of this manuscript including, small molecules (1-20 atoms), large molecules (more than 20 atoms), materials, and catalysis (contains material surfaces with molecules). Each dataset sample contains a 3D atomic structure (positions and atomic numbers) and a set of atomic properties. The atomic properties can be either node-level (e.g., forces) or graph-level (e.g., energy). The datasets are summarized in Table 1, with additional information, including details on train, validation, and test splits, in Appendix H.

To study the ability of pre-trained models to generalize across domains and tasks, we only pre-train on small molecule and catalysis datasets, and fine-tune on small molecule, large molecule, and materials datasets. Our pre-training datasets include the ANI-1x (Smith et al., 2020) and Transition-1x (Schreiner et al., 2022) small molecule datasets and the OC20 (Chanussot et al., 2021) and OC22 (Tran et al., 2022) catalysis datasets. These datasets were chosen due to their diversity and large size.

The combined pre-training dataset contains over 120M training examples with energy and force labels, with the majority of the data ($> 99\%$) coming from non-equilibrium structures. Due to the difference in underlying DFT theory and software used across the datasets, we utilize different prediction heads for each dataset. We also use a per-dataset linear referencing scheme for the energies. For fine-tuning, we use smaller datasets from three domains to evaluate how pre-trained models perform in similar (small molecule) and unseen domains (large molecule and materials). These datasets may contain in-distribution (ID) (i.e., energies and forces) or out-of-distribution (OOD) labels (e.g., QM9's $\Delta_\epsilon$).

| Dataset | Domain | Labels | Elements | Avg size | Train Set | Description |
|---|---|---|---|---|---|---|
| **Pretraining Datasets** | | | | | | |
| OC20 | Catalyst | E, F | 55 | $\sim 73$ (7-225) | 100M | Catalyst relaxations |
| OC22 | Catalyst | E, F | 51 | $\sim 80$ (17-228) | 8M | Oxide catalyst relaxations |
| ANI-1x | Small Molecule | E, F | H, C, N, O | $\sim 15$ (4-63) | 2M | MD simulations |
| Transition-1x | Small Molecule | E, F | H, C, N, O | $\sim 14$ (4-23) | 10M | Reactions database |
| **Finetuning Datasets** | | | | | | |
| Matbench | Materials (OOD) | ID / OOD | 84 | $\sim 30$ (4-444) | $\sim 600$–130k | Material properties |
| QMOF | Materials (OOD) | OOD | 77 | $\sim 109$ (17, 500) | 10k | MOF properties |
| MD17 | Small Mols. (ID) | ID | H, C, N, O | $\sim 13$ (9-21) | 1k | MD simulation |
| QM9 | Small Mols. (ID) | ID / OOD | H, C, N, O | $\sim 18$ (3-29) | $\sim 130$k | QM properties |
| SPICE | Large Mols. (OOD) | ID | H, C, N, O, S | $\sim 46$ (26-96) | 1300, $\sim 34$k | MD simulations |
| MD22 | Large Mols. (OOD) | ID | H, C, N, O | $\sim 67$ (42-370) | $\sim 600$–8k | MD simulations |

Table 1: Summary of datasets and their properties, including the domain, target labels, atomic elements present, their sizes and a brief description.

# 4 JOINT MULTI-DOMAIN PRE-TRAINING

Joint Multi-domain Pre-training (JMP), shown in Figure 1, is based on the intuition that pre-training on a diverse set of chemical domains should lead to better representation learning and thus better generalization through fine-tuning. The pre-training task is framed as a multi-task supervised learning problem, where each label of each pre-training dataset is treated as a separate task. This allows us to pre-train a single backbone model on multiple chemical domains and labels simultaneously.

**Notation:** We use the following notation throughout this section. Let $D = \{D_1, \ldots, D_M\}$ be the set of $M$ datasets that we pre-train on. Each dataset, $D_i$, is a set systems (e.g., molecules or crystals), where each system is a tuple of atomic numbers ($Z$), atomic positions ($R$), and target (i.e., ground-truth) energy ($\hat{E}$) and forces ($\hat{F}$). For a given mini-batch of $B$ systems, $W_b$ is the index of the dataset that system $b \in B$ belongs to, and $N_b$ is the number of atoms in system $b$.

**Model Architecture:** Our goal in this work is to design model-agnostic strategies for supervised pre-training. For our backbone model architecture, we chose GemNet-OC (Gasteiger et al., 2022) for its effectiveness across a wide spectrum of chemical domains as well as at large scales (Sriram et al., 2022). GemNet-OC is a message-passing neural network that computes a node representation $\mathbf{h}_i$ for each atom $i$ and an edge representation $\mathbf{m}_{ij}$ for pairs of nearby atoms, $i$ and $j$. Using these representations, prediction heads compute desired target properties. System-level scalar predictions, such as energy, are computed by summing the node representations, $E = \sum_{i=1}^{N} \mathbf{MLP}(h_i)$. Node-level vector predictions, such as forces, are computed by summing the edge direction unit vectors, weighted by the edge representations, $F_i = \sum_{j=1}^{N} (\mathbf{MLP}(m_{ij}) \cdot \hat{\mathbf{r}}_{ij})$. During pre-training, we compute forces using a direct equivariant block, similar to Klicpera et al. (2021)'s model setup for the OC20 dataset. This is for two reasons: (1) direct force prediction is much more computationally efficient than gradient-based force prediction, as the latter needs to perform a secondary backward pass to compute the gradient of the energy with respect to the atomic positions and (2) previous works (Gasteiger et al., 2022) have shown that for larger datasets, direct force prediction shows much faster convergence while producing similar converged accuracies to gradient-based force prediction.

## 4.1 MULTI-TASK PRE-TRAINING

In the multi-task setting, each dataset has its own energy and force prediction heads, as shown in Figure 1 (left). This allows us to train a single model on multiple datasets simultaneously. In the following sections, we describe each of these imbalances and our proposed solutions in detail.

**Data Normalization:** When pre-training on multiple datasets, we first need to normalize the targets to make sure they are on a common scale across datasets. Since our pre-training task is energy and force prediction, for each dataset we first linearly reference the total energies and then normalize them to unit Gaussian distributions. We normalize the forces by dividing them by component-wise RMS force. This puts the energy and forces for each dataset on a common scale.

**Dataset Size Imbalance:** Our pre-training datasets vary greatly in size, from 2 million to 100 million training samples, for a total of 120M samples. To maintain a proper balance between the total contribution of large, high-resource and small, low-resource pre-training datasets and to prevent overfitting to high-resource datasets and underfitting on low-resource datasets, we use temperature

sampling (Devlin et al., 2018) during batch construction. Specifically, we sample each dataset $i$ with probability $p_i \propto \left(\frac{|D_i|}{\sum_j |D_j|}\right)^{1/T}$, where $|D_i|$ is the number of samples in dataset $i$ and $T$ is the temperature hyperparameter. Inspired by Shaham et al. (2023), which shows that $T = 2$ optimizes model performance on high and low-resource languages for large models, we use $T = 2$.

**System Size Imbalance:** The number of atoms per system varies greatly across our pre-training datasets. For example, Transition-1x has 14 atoms per system on average, while OC22 has 80 atoms per system on average. The naive loss reduction method shown in the non-teal terms of Equation (1), which is the default behavior of most machine learning libraries, computes an atom-level force loss and then averages the force loss across all atoms in the batch. This leads to datasets with more atoms per system dominating the force loss. To address this issue, we propose a structure-wise loss reduction strategy which first computes the average force loss for each system and then computes the average force loss across all systems. This ensures that the relative importance of the force loss is roughly equal across datasets, regardless of the number of nodes per system. In Equation (1), the updates to the naive formulation of the loss function are shown in teal and removed terms are red. This simple change leads to a significant improvement in model performance, as shown in Section 5.1.

$$\mathcal{L} = \underbrace{\frac{1}{B}\sum_{b=0}^{B}\left[\lambda_E^{(W_b)}\left|\hat{E}_b - E_b\right|\right]}_{\text{Energy Loss }(\mathcal{L}_E)} + \underbrace{\frac{1}{B}\frac{1}{\sum_b N_b}\sum_{b=0}^{B}\left[\frac{1}{N_b}\lambda_F^{(W_b)}\sum_{i=0}^{N_b}\left\|\hat{F}_{b,i} - F_{b,i}\right\|_2\right]}_{\text{Force Loss }(\mathcal{L}_F)} \tag{1}$$

**Loss Imbalance Within a Single Dataset:** In the single-dataset setting, $\lambda_E$ and $\lambda_F$ are typically tuned by grid search, but this approach is not feasible in the multi-dataset setting, as there $2 \cdot M$ hyperparameters to tune, and changing one hyperparameter affects the optimal values of the others. Therefore, we need a simple heuristic to determine the loss coefficients for each dataset that provides a reasonable balance between the energy and force losses. Inspired by Tran et al. (2022)'s size invariant force loss, which computes a dynamic $\lambda_F$ based on the number of atoms in each system of the input batch, we fix $\lambda_E^{(i)} = 1$ and $\lambda_F^{(i)} = \langle N \rangle_{D_i}$, where $\langle N \rangle_{D_i}$ is the average number of atoms per system in the $i$th dataset, $D_i$. This provides a reasonable balance between energy and force loss within each dataset.

**Fine-Tuning:** Once we have a fully pre-trained model, we can fine-tune it on downstream tasks. Our fine-tuning procedure is very similar to other fine-tuning procedures in the machine learning literature (Devlin et al., 2018; Zaidi et al., 2022): We discard the pre-training prediction heads, add new randomly initialized prediction heads for the downstream task, and fine-tune the entire model on the downstream task. This procedure is illustrated in Figure 1 (middle). For fine-tuning tasks with force labels, we have the option of using the directly computed forces (i.e., using the direct equivariant block) or computing the forces by taking the gradient of the energy with respect to the atomic positions. Our initial experiments showed that JMP works well with both methods. In our evaluations, however, we chose to compute forces conservatively by taking the gradient of the energy with respect to the atomic positions, as this is the standard approach in the literature.

## 5 EXPERIMENTS

We benchmark our pre-trained models on a diverse set of atomic ML tasks. In previous related works, evaluations are commonly restricted to downstream datasets that align closely with the pre-training dataset's domain (Zaidi et al., 2022; Zhou et al., 2023a). We posit that true success in pre-training models for atomic machine learning tasks requires adeptness at out-of-domain extrapolation. To test this hypothesis, our benchmark uniquely spans across diverse domains including small molecules (QM9 and rMD17), large molecules (MD22 and SPICE), and materials (MatBench and QMOF).

We compare our fine-tuned models (JMP) to randomly initialized models trained from scratch (GN-OC) to demonstrate the effectiveness of JMP. We also compare to previous state-of-the-art models where available. For each task, we present results for both a small ($\sim$30M parameters, labeled with the **-S** suffix) and large ($\sim$230M parameters, labeled with the **-L** suffix) pre-trained model to probe the impact of the model size. These two variants utilize the GN-OC Base and Large backbone architectures from Gasteiger et al. (2022), respectively. Finally, we conduct ablation studies to understand the impact of various components of JMP. More information on the datasets used for

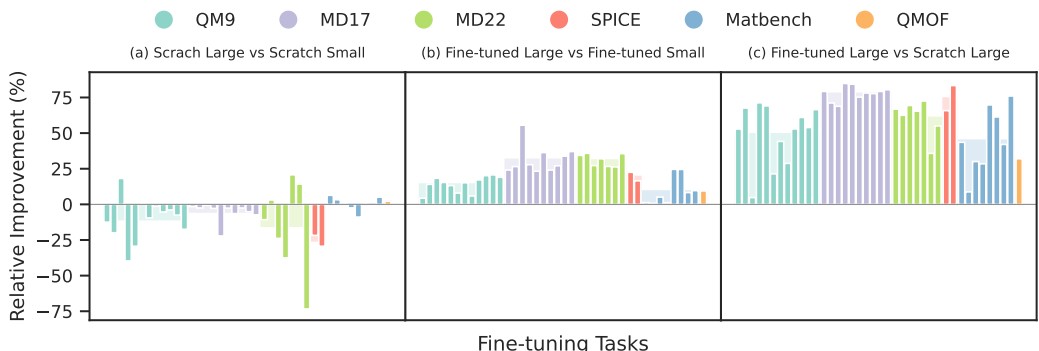

Figure 2: Relative performance improvement across all tasks of all fine-tuning datasets, in percentages, of (a) Scratch Large (**GN-OC-L**) over Scratch Small (**GN-OC-S**), (b) Fine-tuned Large (**JMP-L**) over Fine-tuned Small (**JMP-S**), and (c) Fine-tuned Large (**JMP-L**) over Scratch Large (**GN-OC-L**). GN-OC shows poor scaling to large models, a clear sign of overfitting, whereasJMP reverses this, exhibiting much improved scaling dynamics. JMP also consistently outperforms GN-OC across all domains, datasets, and targets. The shaded rectangles indicate the average relative performance across all tasks for each dataset. The exact percentages can be found in Appendix C.1

pre-training and fine-tuning can be found in Section 3. Details on the pre-training and fine-tuning setup, such as the optimizers, learning rate schedules, and early stopping information, can be found in Appendix F. Exact hyperparameters can be found in Appendix J.

**Common Observations:** We begin by highlighting some common observations across all experiments. First, when training from scratch, GN-OC-L performs 8% worse on average than GN-OC-S, as shown in Figure 2 (a). This is a clear indication of overfitting and has been consistently observed in low-data regimes (Gasteiger et al., 2022). Second, this problem of overfitting is nearly eliminated by JMP, illustrated in Figure 2 (b). On average, JMP-L exhibits an impressive 21% relative performance gain over JMP-S. This indicates that the JMP training procedure is able to effectively leverage the additional capacity of the large model, even in low-data regimes. Third, we observe that pre-training with JMP elevates performance across all domains, datasets, and tasks (Figure 2 (c)), with an average relative improvement of 59% for JMP-L over GN-OC-L.

**Results on Small Molecules - QM9 and rMD17:** For each target of QM9 (Wu et al., 2018), we fine-tune a dedicated model using a simple prediction head with sum pooling for all targets. For $R^2$, we use the same prediction head formulation as Thölke and De Fabritiis (2022). Our results can be found in Table 2 compared against previous state-of-the-art works (Liao and Smidt, 2022; Batatia et al., 2022; Musaelian et al., 2023; Feng et al., 2023a;

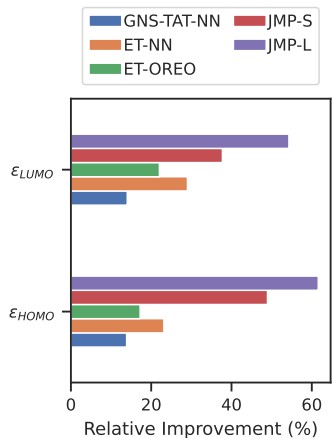

Figure 3: Relative improvement, over training from scratch, of different pre-training methods on QM9's $\epsilon_{LUMO}$ and $\epsilon_{HOMO}$.

Zaidi et al., 2022). With the sole exception of $R^2$, our JMP-L model achieves state-of-the-art results on all QM9 targets. For the $R^2$ target, a similar phenomenon has been observed in previous pre-training works (Zaidi et al., 2022) where the benefits of using pre-trained models are not as pronounced.

In addition to their impressive performance, our JMP-S and JMP-L models demonstrate a large improvement relative to their scratch-trained counterparts. Figure 3 compares this relative improvement — measured on the $\epsilon_{LUMO}$ and $\epsilon_{HOMO}$ targets — to other state-of-the-art pre-training and transfer learning methods for QM9. As shown, JMP outperforms all previous methods by a significant margin. This is a strong signal that our pre-training approach is effective at learning generalizable representations for small molecules. We also report additional pretraining comparisons on all our finetuning benchmarks with a pre-trained model from (Zaidi et al., 2022) and demonstrate significant improvements on all tasks in Appendix A.

**Data overlap:** Due to the limited complexity of small molecules, there is some data overlap between our pre-training datasets (ANI-1x and Transition-1x) and QM9. To check the impact of this overlap

on our results, we evaluate the fine-tuning performance of our JMP-L on a QM9 dataset that excludes the overlapping molecules. Using molecular compositions to identify overlaps, we observe that the exclusion of overlapping molecules has a negligible impact on our results (see Appendix I).

| Target (Units) | TorchMD-Net | Equi-former | MACE | Allegro | Pretrained ET-OREO | Pretrained GNS+TAT+NN | GN-OC-S | GN-OC-L | JMP-S | JMP-L |
|---|---|---|---|---|---|---|---|---|---|---|
| $\mu$ (D) | 0.011 | 0.011 | 0.015 | - | - | 0.016 | 0.020 | 0.023 | 0.010 | **0.008** |
| $\alpha$ ($a_0^3$) | 0.059 | 0.046 | 0.038 | - | - | 0.040 | 0.052 | 0.056 | 0.037 | **0.032** |
| $\varepsilon_{HOMO}$ ($meV$) | 20.3 | 15.0 | 22.0 | - | 16.8 | 14.9 | 21.8 | 22.7 | 11.1 | **8.8** |
| $\varepsilon_{LUMO}$ ($meV$) | 18.6 | 14.0 | 19.0 | - | 14.5 | 14.7 | 17.3 | 18.6 | 10.8 | **8.6** |
| $\Delta\varepsilon$ ($meV$) | 36.1 | 30.0 | 42.0 | - | 26.4 | 22.0 | 38.5 | 40.6 | 23.1 | **19.1** |
| $R^2$ ($a_0^2$) | **0.033** | 0.251 | 0.210 | - | - | 0.440 | 0.210 | 0.171 | 0.200 | 0.163 |
| ZPVE ($meV$) | 1.8 | 1.3 | 1.2 | - | - | 1.0 | 1.2 | 1.2 | 1.0 | **0.9** |
| $U_0$ ($meV$) | 6.2 | 6.6 | 4.1 | 4.7 | - | 5.8 | 7.2 | 9.4 | 3.3 | **2.9** |
| $U$ ($meV$) | 6.4 | 6.7 | 4.1 | 4.4 | - | 5.8 | 6.9 | 9.7 | 3.3 | **2.8** |
| $H$ ($meV$) | 6.2 | 6.6 | 4.7 | 4.4 | - | 5.8 | 7.3 | 8.7 | 3.3 | **2.8** |
| $G$ ($meV$) | 8.3 | 7.6 | 5.5 | 5.7 | - | 6.9 | 8.1 | 9.2 | 4.5 | **4.3** |
| $C_\nu$ (Cal/MolK) | 0.026 | 0.023 | 0.021 | - | - | 0.020 | 0.024 | 0.024 | 0.018 | **0.017** |

Table 2: MAE test split results on all targets of the QM9 dataset. SOTA results are bolded.

For rMD17, we compute forces by taking the negative gradient of the energy with respect to the atomic positions. Table 3 shows our force prediction results on the rMD17 dataset. Similarly to QM9, we observe that JMP consistently outperforms GN-OC across all rMD17 targets. Our JMP-L model achieves state of the art performance in 5 molecules and is very competitive on the rest. Appendix B.1 also shows that JMP achieves SOTA on 6/10 targets on the few-shot 50-sample subset of rMD17.

| Molecules | MACE | Allegro | GN-OC-S | GN-OC-L | JMP-S | JMP-L |
|---|---|---|---|---|---|---|
| Aspirin | 6.6 | 7.3 | 24.3 | 24.7 | 6.7 | **5.1** |
| Benzene | 0.3 | **0.2** | 1.0 | 1.0 | 0.7 | 0.3 |
| Ethanol | 2.1 | 2.1 | 13.0 | 13.3 | 2.8 | **2.0** |
| Malonaldehyde | 4.1 | 4.1 | 21.1 | 25.7 | 5.3 | **4.0** |
| Naphthalene | 1.6 | **0.9** | 5.6 | 5.7 | 2.2 | 1.4 |
| Salicylic acid | **3.1** | 2.9 | 14.7 | 15.1 | 4.6 | 3.4 |
| Toluene | **1.5** | 1.8 | 6.8 | 7.2 | 2.3 | **1.5** |
| Uracil | 2.1 | **1.8** | 12.0 | 12.9 | 4.0 | 2.5 |
| Paracetamol | 4.8 | 4.9 | 17.3 | 18.4 | 5.3 | **4.0** |
| Azobenzene | 3.0 | **2.6** | 11.1 | 11.4 | 4.5 | 3.3 |

Table 3: Force MAE results in meV/Å on the test split of the rMD17 dataset. SOTA is bolded.

**Results on Materials - MatBench and QMOF:** In the materials domain, we fine-tune on the MatBench (Dunn et al., 2020) and QMOF datasets (Rosen et al., 2021). For MatBench, we evaluated all regression tasks that utilize a 3D structure as an input and compared them with competitive models on the leaderboard (De Breuck et al., 2021; Ruff et al., 2023). For QMOF, we predict the band gap target on a 10k split, similarly to Kang et al. (2022); Cao et al. (2023). We use mean pooling for all experiments, except MatBench's phonons, which is the measure of frequency of the highest frequency optical phonon mode peak and thus uses max pooling. Our results can be found in Table 4. We observe that JMP-L achieves SOTA performance across QMOF and on all MatBench tasks. These two datasets contain diverse out-of-domain chemical structures (materials) and out-of-domain target labels (i.e., not energies and forces), relative to the pre-training datasets. JMP's impressive performance is yet another positive signal indicating that JMP is learning generalizable representations.

| Materials (Units) | MODNet (fold0 / mean) | coGN (fold0 / mean) | GN-OC-S (fold0) | GN-OC-L (fold0) | JMP-S (fold0 / mean) | JMP-L (fold0 / mean) |
|---|---|---|---|---|---|---|
| JDFT2D ($meV/atom$) | 25.55 / 33.20 | 22.25 / 37.17 | 26.19 | 25.34 | 20.72 / 30.16 | 23.12 / **29.94** |
| Phonons ($cm^-1$) | 34.77 / 34.28 | 32.12 / 29.71 | 93.45 | 88.74 | 26.6 / 22.77 | 21.28 / **20.57** |
| Dielectric ($unitless$) | 0.169 / 0.271 | 0.178 / 0.309 | 0.225 | 0.211 | 0.133 / 0.252 | 0.119 / **0.249** |
| Log GVRH ($log10(GPA)$) | 0.073 / 0.073 | 0.068 / 0.069 | 0.082 | 0.082 | 0.06 / 0.062 | 0.057 / **0.059** |
| Log KVRH ($log10(GPA)$) | 0.054 / 0.055 | 0.052 / 0.054 | 0.061 | 0.063 | 0.044 / 0.046 | 0.045 / **0.045** |
| Perovskites ($eV/unitcell$) | 0.093 / 0.091 | 0.027 / 0.027 | 0.045 | 0.045 | 0.029 / 0.028 | 0.026 / **0.026** |
| MP Gap ($eV$) | 0.215 / 0.220 | 0.153 / 0.156 | 0.228 | 0.235 | 0.119 / 0.121 | 0.089 / **0.091** |
| MP E Form ($meV/atom$) | 40.2 / 44.8 | 17.4 / 17 | 31.4 | 33.1 | 13.6 / 13.3 | 10.3 / **10.1** |
| | **PT CGCNN** | **PT MOFTransformer** | | | | |
| QMOF | 0.28 | 0.27 | 0.25 | 0.24 | 0.18 | **0.16** |

Table 4: MAE test split results on different targets in the materials domain. SOTA is bolded.

**Results on Large Molecules - MD22 and SPICE:** To further investigate the impact of pre-training on unseen domains, we evaluate two large molecule datasets, MD22 (Chmiela et al., 2023) and SPICE (Eastman et al., 2023) and compare our results to the previous state-of-the-art (Kovacs et al., 2023).

| Molecule | sGDML | MACE | Allegro | GN-OC-S | GN-OC-L | JMP-S | JMP-L |
|---|---|---|---|---|---|---|---|
| Ac-Ala3-NHMe | 34.55 | 3.80 | 4.63 | 5.07 | 6.27 | 2.64 | **1.92** |
| DHA | 32.41 | 2.80 | 3.17 | 2.87 | 3.95 | 2.01 | **1.37** |
| Stachyose | 29.24 | 3.80 | 4.21 | 2.22 | 3.85 | 2.69 | **1.73** |
| AT-AT | 29.97 | 4.30 | 4.13 | 5.38 | 5.96 | 3.02 | **1.98** |
| AT-AT-CG-CG | 30.48 | 5.00 | 5.55 | 5.80 | 5.62 | 3.28 | **2.11** |
| Buckyball Catcher | 29.57 | 3.70 | - | 10.35 | 8.20 | 3.08 | **2.26** |
| Double Walled Nanotubes | 22.68 | 12.00 | - | 11.20 | 9.61 | 8.36 | **6.17** |
| Solvated Amino Acids | | | | 6.20 | 8.02 | 1.60 | **1.33** |
| Dipeptides | | | | 2.46 | 2.99 | 1.32 | **1.02** |

Table 5: Force MAE results in meV/Å on test splits of large molecule datasets. SOTA is bolded.

For SPICE, we only use the large molecule sub-tasks: solvated amino acids and dipeptides. Similar to rMD17, we compute forces by taking the negative gradient of the energy with respect to the atomic positions. However, for MD22's Buckyball Catcher and Double-Walled Nanotubes, we were unable to fit these large structures in memory with using gradient-based force predictions; therefore, we used direct force prediction heads instead. Our results can be found in Table 5. Once again, our model demonstrates state-of-the-art results across all molecules of MD22 and all tasks of SPICE.

## 5.1 Ablation Studies

Our ablations demonstrate the impact of various changes to JMP on the downstream fine-tuning performance. We performed pre-training experiments including dataset sampling strategies, loss formulation, and regularization strategies, and observed their impact on fine-tuning. Given the computational cost of training models on the full pre-training dataset, ablation experiments were conducted on a scaled-down version of the full pre-training dataset containing a randomly selected $\sim$2.5M examples. All pre-training models are trained for 10 epochs. Similarly, fine-tuning for these experiments was run on only one task from each of the fine-tuning datasets (MD17: Aspirin, MD22: Stachyose, QM9: $\Delta\epsilon$, MatBench: MP E Form, QMOF: Band Gap, and SPICE: Solvated Amino Acids). Additional ablations, including using fully balanced ($T = \infty$) sampling, threshold regression loss for energies and forces, and automatic task weighting strategies such as PCGrad (Yu et al., 2020) are explored in Appendix B. Table 6 shows the mean improvement, relative to the base, across all the fine-tuning tasks described above. A summarized insight of each ablation study follows:

**Base (B):** Base refers to the naive implementation of a multi-task pre-training model without temperature sampling, structure-wise loss reduction, or additional regularization. This model serves as the baseline for comparison.
**Temperature Sampling:** Temperature-based sampling with $T = 2$ provides a moderate improvement, while higher values (e.g., $T = \infty$) show diminishing returns. This is consistent with Shaham et al. (2023), which shows that for large-enough models, $T = 2$ provides ideal performance across both low and high resource datasets.
**Structure-Wise Loss Reduction (SWL):** The application of the structure-wise loss reduction strategy proved to be a substantial improvement on the model's performance, with **T2 + SWL** offering a 7.7% improvement over **B**.
**Weight Decay (WD):** Elevating the weight decay regularization parameter to 0.1 (from the default 0.01) brings the collective improvement to 11.4% over **B**.

| Ablations | $\mathbb{E}[\mathbf{RI}](\%)$ |
|---|---|
| Base (Temperature 1.0) [**B**] | 0% |
| B + Temperature 2.0 [**T2**] | 2.2% |
| B + Temperature $\infty$ [**T$\infty$**] | 2.6% |
| T2 + SW Loss Averaging [**SWL**] | 7.7% |
| SWL + Weight Decay [**WD**] | 11.4% |
| SWL + Dropout [**DO**] | 11.4% |
| WD + Edge Dropout [**ED**] | 13.2% |
| WD + ED + EMA Weights [**EMA**] | 12.4% |
| EMA + OC20 Only [**OC20**] | -9.9% |

Table 6: Ablation results demonstrating the mean relative improvements of each method relative to the base method (**B**), averaged over ablation subsplits.

**Dropout (DO):** Using dropout with $p = 0.1$ on atom update layers yielded similar uplift to **WD**.
**Edge Dropout (ED):** For this ablation, we drop $p = 0.1$ of the edges at every step. We then scale the embeddings of remaining edges by a factor of $\frac{1}{1-p}$. This yielded a small improvement over **WD**, increasing the collective improvement to 13.2% over **B**.
**Exponential Moving Average (EMA):** Fine-tuning on EMA weights did not improve performance.
**OC20 Only (OC20):** To understand the impact of multi-task pre-training, we trained a model on the OC20 dataset only. We selected a 120M subset of OC20 to match the number of examples in the full JMP pre-training dataset. Note that this means that the dataset used in the **OC20** ablation contains $48\times$ more data points than the rest of our ablations. Despite this, **OC20** performed substantially worse than **B**, indicating that diverse multi-task pre-training is important for generalization.

Based on our ablation results the two most important changes from **B** were **SWL** and regularization methods like **WD**, **DO**, and **ED**. These results are consistent with Kurin et al. (2022), which demonstrates the effectiveness of regularization in multi-task learning. Our final model integrates temperature-based sampling (**T2**), structure-wise loss reduction strategy (**SWL**), an amplified weight decay regularization parameter of 0.1 (**WD**), edge dropout with $p = 0.1$ (**ED**), and **EMA** despite not showing a performance boost as it has been standard for training GemNet (Gasteiger et al., 2022).

## 5.2 Computational Cost Analysis

Pre-training JMP-L required significant computational resources, which is typical for foundation model approaches. We pre-trained JMP-L on 128x V100 32GB GPUs for 2 epochs, which took around 34,400 GPU hours in total (see Appendix G for exact training times and $CO_2$ impact). While this is a substantial upfront investment, it enables efficient fine-tuning across a diverse set of downstream tasks.

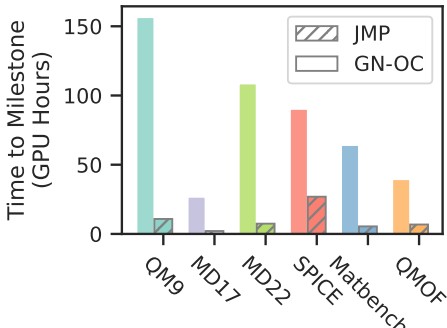

Figure 4: The number of GPU hours, averaged for each dataset, required to train GN-OC-L to convergence and to fine-tune JMP-L to match GN-OC-L's performance. Overall, fine-tuning JMP-L was able to match GN-OC-L's performance in $\frac{1}{12}$ the time.

We evaluated JMP-L fine-tuning performance versus training models from scratch (i.e., GN-OC-L). Training GN-OC-L on the downstream tasks until convergence based on our stopping criteria took around 3,300 GPU hours in total across all tasks. In contrast, fine-tuning JMP-L on the same tasks took only around 275 GPU hours total to match the performance of the models trained from scratch. This 12x reduction in compute demonstrates the significant benefits of pre-training. Figure 4 shows this difference in compute requirements, averaged for each fine-tuning dataset.

## 6 Conclusion

Our findings demonstrate the promise of pre-training strategies that leverage diverse data to improve atomic property prediction. Our source code and pre-trained models are publicly available on GitHub[1]. We hope this will spark further research and innovation in this area to accelerate progress towards a foundation model for chemistry. It is essential to acknowledge the potential for bias in our models and datasets, which can impact their prediction capability and quality. Additionally, given the potential for misuse of atomic machine learning models, we emphasize that these models must be used responsibly.

There are several limitations of this study that provide fertile ground for future research. Due to the large computation expense of pre-training new models, our study only experimented with the GemNet-OC backbone model. Further exploration of different backbone models is warranted. Additionally, the current methodology of discarding pre-training prediction heads before fine-tuning may not be optimal, particularly for datasets with similar labels to the pre-training sets. Lastly, the model size used in this study, though substantial, is dwarfed by the largest models in NLP and CV. We anticipate that employing larger models, trained with the help of recent data and model parallelism techniques (Sriram et al., 2022), could enhance performance.

In summary, this work presents Joint Multi-domain Pre-training (JMP), a novel atomic pre-training strategy that leverages diverse atomic datasets to learn rich representations through multi-task regression. By effectively formulating and regularizing the multi-task learning problem, we achieve remarkable performance on various in-domain and out-of-domain fine-tuning tasks. Scaling up to a large model with over 235 million parameters leads to improved performance on all downstream tasks, even low-resource ones like rMD17. This suggests that pre-training not only enhances accuracy but also facilitates effective scaling, allowing models to benefit from increased capacity without overfitting. Additionally, we establish a comprehensive set of fine-tuning benchmarks across various chemical domains and tasks. Building off the results present here to create larger, more general, and more accurate ML potentials will remain significant challenge for the field moving forward.

---

[1]`https://github.com/facebookresearch/JMP`

**Acknowledgements:** We would like to thank Dr. Abhishek Das, who provided valuable insights and feedback on the research direction and manuscript. We would also like to thank Anuroop Sriram and Dr. Johannes Gasteiger for their valuable insight. Finally, we would like to thank the entire FAIR Chemistry team for their support and feedback.

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

APPENDIX

**Table of Contents:**

## A    ADDITIONAL PRETRAINING COMPARISONS

We compare our finetuning results on all benchmarks with the publicly available checkpoint from Zaidi et al. (2022) that performs pretraining via denoising. The only checkpoint available is with TorchMDNet backbone and not their best-performing model, therefore we just add results for pretraining ET+NN where NN stands for noisy nodes as described in the original work. In this method, they pretrain on the PCQM4Mv2 (Hu et al., 2021) dataset which is a 3D small molecule dataset. We perform significantly better on all datasets (including in the small molecule domain) showing the impact of our supervised joint pretraining approach. A model trained on just a small molecule dataset in a denoising setup focused on equilibrium systems performs worse than the model trained on a supervised setup on non-equilibrium combining datasets from multiple chemical domains. Our Figure 3 in the main paper also demonstrated the relative improvement coming from pretraining relative to models performed from scratch.

| Molecules | Pretrained ET+NN | JMP-S | JMP-L |
|---|---|---|---|
| Aspirin | 15.1 | 6.7 | **5.1** |
| Benzene | 1.0 | 0.7 | **0.3** |
| Ethanol | 8.9 | 2.8 | **2.0** |
| Malonaldehyde | 13.0 | 5.3 | **4.0** |
| Naphthalene | 4.5 | 2.2 | **1.4** |
| Salicylic acid | 9.8 | 4.6 | **3.4** |
| Toluene | 4.8 | 2.3 | **1.5** |
| Uracil | 6.4 | 4.0 | **2.5** |
| Paracetamol | 12.6 | 5.3 | **4.0** |
| Azobenzene | 7.6 | 4.5 | **3.3** |

Table 7: Force MAE errors in meV/Å on the test split of the rMD17 dataset.

## B    ADDITIONAL RESULTS AND EXPERIMENTS

### B.1    EXPANDED rMD17 RESULTS: TRAINING ON 50 EXAMPLES

Following the idea presented by Batatia et al. (2022) we reduced the rMD17 per molecule training set size from $1000 \rightarrow 50$ examples and evaluated the performance of the JMP-L model. The test set remained unchanged. Table 10 shows these results. The large pre-trained model provided state-of-the-art performance on 6 out of 10 molecules and gives competitive results on the rest. Most notably, we

| Target (Units) | Pretrained ET+NN | JMP-S | JMP-L |
|---|---|---|---|
| $\mu$ $(D)$ | 0.015 | 0.010 | **0.008** |
| $\alpha$ $(a_0^3)$ | 0.069 | 0.037 | **0.032** |
| $\varepsilon_{\text{HOMO}}$ $(meV)$ | 23.1 | 11.1 | **8.8** |
| $\varepsilon_{\text{LUMO}}$ $(meV)$ | 19.1 | 10.8 | **8.6** |
| $\Delta\varepsilon$ $(meV)$ | 39.8 | 23.1 | **19.1** |
| $R^2$ $(a_0^2)$ | 0.556 | 0.200 | **0.163** |
| ZPVE $(meV)$ | 1.1 | 1.0 | **0.9** |
| $U_0$ $(meV)$ | 6.0 | 3.3 | **2.9** |
| $U$ $(meV)$ | 6.0 | 3.3 | **2.8** |
| $H$ $(meV)$ | 6.1 | 3.3 | **2.8** |
| $G$ $(meV)$ | 6.9 | 4.5 | **4.3** |
| $C_\nu$ (cal/mol K) | 0.021 | 0.018 | **0.017** |

Table 8: Average absolute error results on all targets of the QM9 dataset.

| Molecule | Pretrained ET+NN | JMP-S | JMP-L |
|---|---|---|---|
| Ac-Ala3-NHMe | 8.53 | 2.64 | **1.92** |
| DHA | 7.23 | 2.01 | **1.37** |
| Stachyose | 10.12 | 2.69 | **1.73** |
| AT-AT | 10.35 | 3.02 | **1.98** |
| AT-AT-CG-CG | 8.81 | 3.28 | **2.11** |
| Buckyball Catcher | - | 3.08 | **2.26** |
| Double Walled Nanotubes | - | 8.36 | **6.17** |
| Solvated Amino Acids | 7.11 | 1.60 | **1.33** |
| Dipeptides | 6.56 | 1.32 | **1.02** |

Table 9: Force MAE results in meV/Å on the test splits for large molecule datasets (MD22 and SPICE). Missing numbers are due to runs failing or hitting NaNs.

see that the fine-tuned model performs significantly better than the GN-OC-L model. These results offer preliminary evidence that reasonable few-shot performance may be achievable with a large pre-trained atomic model.

| Molecules | NequIP | MACE | GN-OC-L | JMP-L |
|---|---|---|---|---|
| Aspirin | 52.0 | 43.9 | 119.7 | **36.8** |
| Benzene | 2.9 | **2.7** | 8.8 | 2.8 |
| Ethanol | 40.2 | 32.6 | 95.3 | **22.2** |
| Malonaldehyde | 52.5 | 43.3 | 159.2 | **42.9** |
| Naphthalene | 10.0 | **9.2** | 39.5 | 9.6 |
| Salicylic acid | 35.0 | 28.4 | 108.5 | **26.4** |
| Toluene | 15.1 | **12.1** | 53.3 | 12.4 |
| Uracil | 40.1 | 25.9 | 135.5 | **25.8** |
| Paracetamol | 39.7 | 31.5 | 66.4 | **27.3** |
| Azobenzene | 20.0 | **17.7** | 139.0 | 17.8 |

Table 10: Average absolute force prediction errors in meV/Å on the test split of the revised MD17 dataset comparing the performance of fine-tuning our pre-trained model with GN-OC trained from scratch, as well as other state-of-the-art results. For all runs, the train set was limited to only 50 examples.

## B.2   PDBBind v2013 Binding Affinity Results

We also evaluate the performance of our pre-trained model on the core set of the PDBBind v2013 dataset (Liu et al., 2015), provided in the MoleculeNet benchmark (Wu et al., 2018), which contains 154 protein-ligand complexes. The dataset contains 3D structures of protein-ligand complexes, as well as the binding affinity of the ligand to the protein. Following MoleculeNet, we use a random 80%/10%/10% train/validation/test split, and we report the RMSE metric. Table 11 shows the results of our pre-trained model compared to the state-of-the-art results. We see that our pre-trained model achieves state-of-the-art performance on this dataset, further demonstrating that our pre-trained model can be used for a wide range of molecular tasks, including those that involve protein-ligand complexes.

| Method | $-log\frac{K_d}{K_i}$ **RMSE** |
|---|---|
| **JMP-L** | **1.36** |
| **DeepBindGCN_RG_x** | 1.49 |
| **SE-OnionNet** | 1.69 |
| **DeepBindRG** | 1.81 |
| **GraphBAR (dataset 4, best)** | 1.63 |
| **BAPA** | 1.45 |

Table 11: Binding affinity RMSE results on the PDBBind v2013 dataset.

## B.3   Additional Ablation Studies

**Fully balanced datasets:** Finding the optimal dataset scaling is important for pre-training with datasets of variable size. While we investigated dataset scaling with a temperature of 2.0 in the main ablation results, we additionally tested the impact of fully balancing the datasets (high temperature). The results for this ablation can be found in Table 13, labeled as B + **Fully balanced**. Compared to the base pre-trained model where there was no dataset scaling, the fully balanced run provided a mean relative improvement (RI) of 2.61% on the ablation fine-tuning tasks. This is very similar to the improvement we see with temperature 2.0 (mean RI over base of 2.29%), but it requires more training steps to reach an epoch of the largest dataset. As a result, we used temperature 2.0 dataset scaling for all our models trained on the full pre-training dataset.

**Threshold loss:** DFT is an approximate method and as a result the labels (e.g. energy and forces) in DFT datasets have some error associated with them. In our multi-dataset case, many datasets are computed using different DFT engines and levels of theory, leading to distinct noise distributions for each dataset. To address this, we experiment with a threshold loss which measures the distance between the model's prediction and the ground-truth (i.e., DFT-computed) label, but only penalizes predictions that are outside of a given threshold. To implement this threshold loss, we modify the loss original functions, $\mathcal{L}_i$, by incorporating the predefined physically motivated margins for each dataset as shown in Table 12. The equation for the threshold loss is shown in Equation (2).

$$\mathcal{L}_i(\hat{y}, y) = \begin{cases} \mathcal{L}_i^0(\hat{y}, y) & \text{if } M_i(\hat{y}, y) \geq \text{margin} \\ 0 & \text{otherwise} \end{cases} \tag{2}$$

where $\mathcal{L}_i^0$ is the original loss function for dataset $i$, and $M_i$ is the metric used to compute the margin for dataset $i$. For all our datasets, we use the mean absolute error (MAE) as the metric for computing the margin. Table 13 shows ablation studies for a model which used this threshold loss mechanism, labeled as B + T2 + SWL + **Threshold Loss**. While our initial experiments indicated that the threshold loss was a promising modification, our final ablation results show that the threshold loss hurts performance.

| Dataset | Energy (eV) | Forces (eV/Å) |
|---|---|---|
| ANI-1x | 0.043 | 0.01 |
| Transition-1x | 0.043 | 0.01 |
| OC20 | 0.1 | 0.03 |
| OC22 | 0.1 | 0.03 |

Table 12: Summary of energy and force threshold values for different datasets.

**Automatic Task Weighting with PCGrad:** PCGrad (Yu et al., 2020) is a "gradient surgery" method that attempts to address the optimization challenges in multi-task learning by projecting each task's gradient onto the normal plane of the gradient of any other task that has a conflicting gradient. We evaluate the usage of PCGrad alongside our structure-wise loss reduction strategy. Our experiments showed that PCGrad (**PCG**) does not offer performance improvements over **SWL**. These results are consistent with Kurin et al. (2022), which shows that adequate regularization can mitigate the need for complex multi-task optimization methods.

| Ablations | Mean RI (%) |
|---|---|
| Base (**B**) | 0% |
| B + **Fully balanced** | **2.61%** |
| B + Temperature 2.0 (**T2**) | 2.29% |
| T2 + SW Loss Averaging (**SWL**) | 7.68% |
| SWL + **Threshold Loss** | **5.38%** |
| SWL + PCGrad (**PCG**) | 6.48% |
| SWL + Weight Decay (**WD**) | 11.37% |
| SWL + Dropout (**DO**) | 11.41% |
| SWL + Edge Dropout (**ED**) | 13.18% |
| SWL + WD + EMA Weights (**EMA**) | 12.38% |
| EMA + OC20 Only (**OC20**) | **-9.88 %** |

Table 13: Additional experiments with all ablation results

### B.4 FINE-TUNING LR SCHEDULING

Learning rate scheduling has a large impact on the downstream fine-tuning performance. We utilize with a robust learning rate scheduling strategy that we employ in all our fine-tuning experiments. Namely, we combine the ideas behind **linear warmup**, **cosine decay** (Loshchilov and Hutter, 2016),

**Layerwise Learning Rate Decay (LLRD)** (Howard and Ruder, 2018), and **ReduceLROnPlateau**. Let $\alpha$ be our learning rate, $N_w$ be the number of warmup epochs, $f_w$ be the warmup initial learning rate coefficient, $N_c$ be the number of cosine decay epochs, $f_c$ be the cosine decay final learning rate coefficient, $N_p$ be the number of patience epochs for ReduceLROnPlateau, $f_p$ be the patience learning rate coefficient, and $D_i$ be the LLRD decay coefficient for layer $i$. Then, our learning rate scheduling strategy for layer $i$ is described below:

1. **Linear Warmup**: During the initial phase of training, the learning rate $\alpha$ begins at $f_w \cdot \alpha \cdot D_i$ and gradually escalates to reach $\alpha \cdot D_i$ over $N_w$ warmup epochs. This strategy aids in preventing substantial gradients at the beginning of training, thereby ensuring a stable optimization process.

2. **Cosine Decay**: After the warmup phase, we transition to a cosine decay strategy. The learning rate starts from $\alpha \cdot D_i$ and decays to $f_c \cdot \alpha$ over $N_c$ epochs. This phase facilitates the gradual reduction of the learning rate, enabling efficient model convergence. The final learning rate after the cosine decay phase is the same for all layers, meaning LLRD only influences the warmup and cosine decay phases.

3. **ReduceLROnPlateau**: The final phase is governed by the ReduceLROnPlateau strategy. It commences with the learning rate set at $f_c \cdot \alpha$, and if the validation loss does not decrease for $N_p$ epochs, the learning rate is decreased by multiplying it with $f_p$. This dynamic adjustment of the learning rate based on validation loss performance assists in fine-tuning the model parameters towards the end of the training process.

This scheduling strategy provides a robust method to adjust the learning rate across different phases of the model training, thereby balancing the need for both rapid learning and careful optimization as the model converges to the best solution. Table 14 shows the fine-tuning performance of the **JMP-L** model when fine-tuned on the Aspirin molecule of the rMD17 dataset using different components of our LR scheduling strategy. We observe that the performance of the model improves as we add each component of the LR scheduling strategy. We can see that all components of our scheduling strategy contribute to the performance of the model. Overall, we see a massive improvement of 31% in the force MAE when we use our full LR scheduling strategy when compared to simply using a cosine decay strategy. This demonstrates the importance of a robust learning rate scheduling strategy for fine-tuning the model on downstream tasks. For our final runs, the specific values of the LR scheduling hyperparameters can be found in Appendix J.

| LR Schedule | Forces MAE |
|---|---|
| Warmup + Cos | 7.8 |
| Warmup + Cos + LLRD | 6.4 |
| Warmup + Cos + LLRD + RLP | 5.1 |

Table 14: Impact of learning rate schedules on Aspirin Force MAE (meV/Å) when evaluated on the validation set. Cos: Cosine Decay, LLRD: Layerwise Learning Rate Decay, RLP: ReduceLROn-Plateau

## C   EMBEDDING AND ACTIVATION VISUALIZATIONS

In this section, we illustrate the potency of our pre-trained model's embeddings through visualizations. The t-SNE plot in Figure 5 represents the node-level ($h$) and edge-level ($m$) GN-OC embeddings of the JMP-L model, computed, averaged across the entire system as shown in Equation (3), where $N$ is the number of nodes in the system, $E$ is the number of edges in the system, $h \in (N, D_h)$ is the GN-OC node embedding tensor for the system, and $m \in (E, D_m)$ is the GN-OC edge embedding tensor for the system. We randomly select structures across all pre-training and fine-tuning development datasets (as described in Appendix H.3) and compute these aggregated system-level embeddings. Each data point corresponds to a unique structure, and its color indicates the dataset from which the structure originated.

$$\tilde{h} = \frac{1}{N}\sum_{i=0}^{N} h_i \quad \text{and} \quad \tilde{m} = \frac{1}{E}\sum_{e=0}^{E} m_e \tag{3}$$

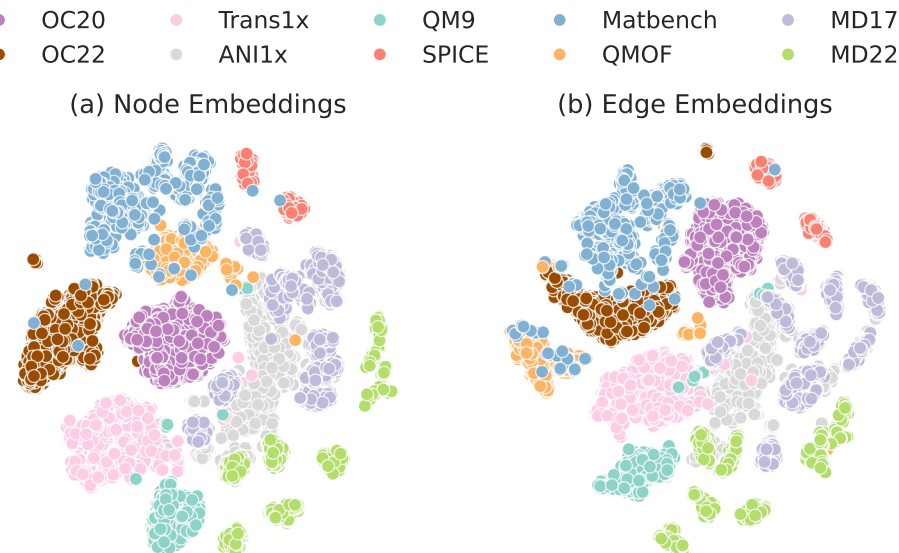

Figure 5: t-SNE visualizations of the node-level ($\tilde{h}$) and edge-level ($\tilde{m}$) JMP-L embeddings for randomly selected structures from all pre-training and fine-tuning development datasets. Each point represents a structure, and the color indicates the dataset from which the structure was sampled.

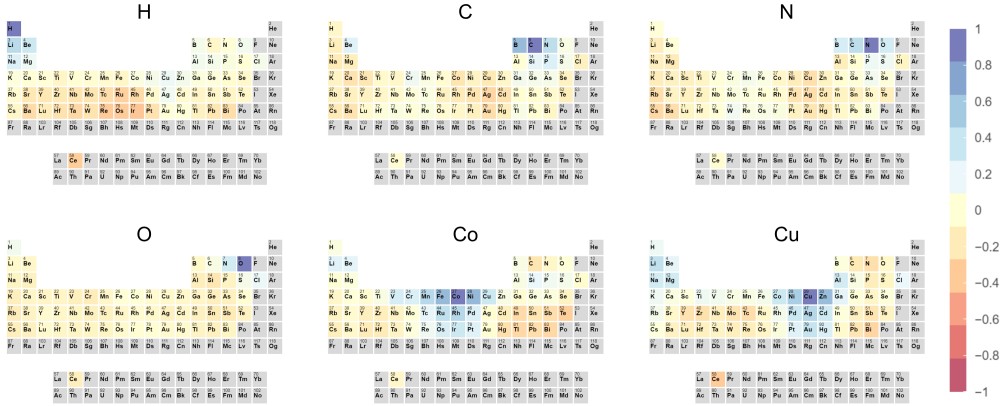

Figure 6: Visualization of the cosine similarity between the learned atom embeddings for H, C, N, O, Co, and Cu with all other elements in the dataset. Note that nearby atoms in the periodic table have more similar embeddings, which is consistent with known element properties.

Furthermore, Figure 6 provides an intriguing visualization of the cosine similarities between the learned atom embeddings for a selection of elements (H, C, N, O, Co, and Cu) with all other elements in the dataset. The plot reveals that atoms located adjacently on the periodic table exhibit more similar embeddings, a result that aligns well with their known elemental properties. This underlines the model's ability to accurately capture and encode fundamental chemical properties within its learned embeddings.

## C.1 EXACT PERCENTAGES FOR FIGURE 2

The per-dataset averages values (with translucent shading) for Figure 2 are: (a) QM9 -10.7%, MD17 -5.4%, MD22 -15.4%, SPICE -25.6%, Matbench 0.4%, QMOF 2.2%; (b) QM9 14.3%, MD17 31.6%, MD22 31.1%, SPICE 19.5%, Matbench 9.4%, QMOF 9.5%; and (c) QM9 49.5%; MD17 77.9%; MD22 61.0%; SPICE 74.6%; Matbench 45.0%; QMOF 32.0%.

# D ADDITIONAL METHOD INFORMATION

## D.1 OPTIMAL TASK WEIGHTING AND REGULARIZATION

The problem of optimal task weighting is an active area of research in multi-task learning. Existing works range from dynamically weighting tasks based on uncertainty estimates computed from their loss (Kendall et al., 2018) to performing "gradient surgery" on per-task gradients to alleviate conflicting gradients. However, recent work by Kurin et al. (2022) has shown that *unitary scalarization*, or simply summing the losses across all tasks, matches or outperforms most existing methods, provided that adequate regularization is used. Inspired by these findings, we use unitary scalarization for pre-training, and we use the following regularization techniques: We use a weight decay of $0.1$, edge dropout (Rong et al., 2019) with $p = 0.1$, and exponential moving average with a decay of $0.99$ on the model weights.

## D.2 FINE-TUNING FORCE COMPUTATION

For fine-tuning tasks with force labels, we have the option of using the directly computed forces (i.e., using the direct equivariant block) or computing the forces by taking the gradient of the energy with respect to the atomic positions. Our initial experiments showed that JMP works well with both methods. In our evaluations, however, we chose to compute forces conservatively by taking the gradient of the energy with respect to the atomic positions, as this is the standard approach in the literature.

# E STRUCTURE-WISE LOSS REDUCTION CODE

In this section, we provide the code for the structure-wise loss reduction (SWL) loss function, as well as the original GemNet-OC loss function for comparison. The code is written in PyTorch, and is shown in Listings 1 and 2. Table 15 below maps the notation used in the sample code to the notation used in Equation (1) in the main text.

```
def gemnet_oc_loss(
    E: Tensor, # (batch_size,)
    F: Tensor, # ((batch_size*num_atoms), 3)
    E_target: Tensor, # (batch_size,)
    F_target: Tensor, # ((batch_size*num_atoms), 3)
    batch_idx: Tensor, # ((batch_size*num_atoms),)
    lambda_E: Tensor, # scalar
    lambda_F: Tensor, # scalar
```

---

[2] The `batch_idx` tensor is a product of PyTorch Geometric's sparse graph batching, which constructs a batched graph as one large graph with disconnected components. The `batch_idx` tensor maps each node in the batched graph to its corresponding component in the original batch. In the notation of the paper, we do not use this explicitly to simplify the notation. Instead, we use subscripts on node-level quantities to denote the component that the node belongs to. For example, $F_{b,i}$ denotes the force vector of the $i$-th atom in the $b$-th component of the batched graph, or $N_b$ denotes the number of atoms in the $b$-th component of the batched graph.

| Sample Code Notation | Paper Notation |
|---|---|
| E | $E$ |
| F | $F$ |
| E_target | $\hat{E}$ |
| F_target | $\hat{F}$ |
| lambda_E | $\lambda_E$ |
| lambda_F | $\lambda_F$ |
| batch_idx | Not present[2] |
| dataset_idx | $W$ |

Table 15: Mapping of notation used in the sample code to the notation used in Equation (1) in the main text.

```
9  ):
10      # Energy loss computation
11      systemwise_energy_loss = (E - E_target) ** 2 # (batch_size,)
12      energy_loss = lambda_E * torch.mean(
13          systemwise_energy_loss
14      ) # scalar
15
16      # Force loss computation
17      atomwise_F_loss = torch.norm(
18          F - F_target,
19          p=2,
20          dim=-1
21      ) # (batch_size*num_atoms,)
22      force_loss = lambda_F * torch.mean(atomwise_F_loss) # scalar
23
24      return energy_loss + force_loss
```

Listing 1: Original GemNet-OC loss function

```
1  def jmp_loss(
2      E: Tensor, # (batch_size,)
3      F: Tensor, # ((batch_size*num_atoms), 3)
4      E_target: Tensor, # (batch_size,)
5      F_target: Tensor, # ((batch_size*num_atoms), 3)
6      batch_idx: Tensor, # ((batch_size*num_atoms),)
7      dataset_idx: Tensor, # (batch_size,)
8      lambda_E: Tensor, # (num_datasets,)
9      lambda_F: Tensor, # (num_datasets,)
10  ):
11      # Energy loss computation
12      # Energy computation is the same as the
13      #    original GemNet-OC loss function,
14      #    with the only difference being that
15      #    we have task-specific energy loss coefficients.
16      systemwise_energy_loss = (E - E_target) ** 2 # (batch_size,)
17      energy_loss = torch.mean(
18          systemwise_energy_loss
19          * lambda_E[dataset_idx]
20      ) # scalar
21
22      # Force loss computation
23      atomwise_F_loss = torch.norm(
24          F - F_target,
25          p=2,
26          dim=-1
27      ) # (batch_size*num_atoms,)
28      # The only difference is that we now
29      #    perform a structure-level averaging,
```

```
30      #   instead of an atom-level averaging.
31      structurewise_force_loss = scatter_mean(
32          atomwise_F_loss,
33          batch_idx
34      ) # (batch_size,)
35      # Similarly to the energy loss, we now have
36      #   task-specific force loss coefficients.
37      force_loss = torch.mean(
38          structurewise_force_loss
39          * lambda_F[dataset_idx]
40      ) # scalar
41
42      return energy_loss + force_loss
```

Listing 2: Our proposed loss function

## F    TRAINING SETUP

### F.1    PRE-TRAINING

**Optimizer** We use the AdamW (Kingma and Ba, 2014) optimizer with a learning rate of $0.0003$, $\beta_1 = 0.9$, $\beta_2 = 0.95$, and a weight decay value of $0.1$.
**Learning Rate Scheduling**. During pre-training, we use a linear-warmup with cosine decay learning rate schedule. The linear warmup starts with $0.2 \cdot$ LR and warms up to LR over 2000 steps. The cosine decay reduces the learning rate to $0.1 \cdot$ LR over 2 epochs.

### F.2    FINE-TUNING

**Optimizer** We use the AdamW (Kingma and Ba, 2014) optimizer with a learning rate of $0.00008$, $\beta_1 = 0.9$, $\beta_2 = 0.95$, and a weight decay value of $0.1$.
**Loss Function**. We use the MAE loss function for scalar targets (e.g., energy, band-gap) and the L2 distance loss function vector targets (e.g., forces).
**Learning Rate Scheduling**. Across all fine-tuning tasks, we utilize the following learning rate schedule: (1) Warmup over 5 epochs, (2) cosine decay over 32 epochs, and (3) reduce on plateau for the remainder of training. We also utilize Layer-wise Learning Rate Decay (LLRD) (Howard and Ruder, 2018) during phases (1) and (2). LLRD employs higher initial learning rates for final, more task-specific layers. The final learning rate after phase (2), however, is the same for all layers. See Appendix B.4 for more information on the fine-tuning LR scheduling.
**Early Stopping**. All our fine-tuning runs use the following stopping criteria: (1) Early stopping with a patience of 50 epochs, (2) maximum of 500 epochs or 7 days of training, or (3) the learning rate dropping below $10^{-8}$. For the rMD17 dataset, due to the small size of the train set, we use a patience of 1000 and a maximum of 100,000 epochs, similar to Musaelian et al. (2023).

## G    MODEL TRAINING TIMES AND $CO_2$ IMPACT

Table 16 shows the total training time (in GPU-hours) of pre-training our model on the pre-training datasets, fine-tuning the model on all fine-tuning datasets, and training baseline scratch models for each fine-tuning dataset. The substantial computational resources required for pre-training the models cannot be understated. These numbers, however, are significantly offset when considering the subsequent fine-tuning phase. Remarkably, the fine-tuned models, starting from the pre-trained checkpoints, demonstrated superior performance while utilizing approximately half of the computational resources required by the models trained from scratch. This efficacious use of resources when applying our technique substantiates its viability. Moreover, the initial investment of resources in pre-training the models is well compensated as these models, once trained, can be reused across multiple applications, thereby amplifying their utility and cost-effectiveness.

All experiments were conducted using private infrastructure, which has a carbon efficiency of 0.432 $kgCO_2eq/kWh$. A cumulative of 46400 hours of computation was performed on hardware of type Tesla V100-SXM2-32GB (TDP of 300W). Total emissions are estimated to be 6013.44 $kgCO_2eq$ of

which 100 percents were directly offset. Estimations were conducted using the Machine Learning Impact calculator presented in Lacoste et al. (2019).

| Model | Time (GPU-hours) | $CO_2$ Emissions |
|---|---|---|
| JMP-L Pre-Training | 34400 | 4458.24 |
| JMP-S Pre-Training | 5700 | 738.72 |
| JMP-L Fine-Tuning | 3000 | 388.8 |
| Scratch GN-OC-L Training | 3300 | 427.68 |

Table 16: Training times of different models in this work and their corresponding $CO_2$ emission estimates

## H  ADDITIONAL DATASET DETAILS

### H.1  PRE-TRAINING DATASETS

For the purpose of pre-training, we selected two datasets from the catalysis domain, namely OC20 and OC22, as well as two small molecules datasets, ANI-1x and Transion-1x. These datasets were chosen due to their substantial training sizes and the diversity of structures they offer. Note that all of the pre-training datasets we used contain energy and forces labels.

**Linear Referencing and Normalization:** The underlying DFT functional and DFT engine (e.g. VASP vs. Orca) used differs between our pre-training datasets, resulting in variations in energy magnitudes across them. To address these differences and establish a consistent energy reference, we first compute an element specific energy reference for each of these datasets independently. This is achieved by calculating a linear reference across the training split of each dataset to find the per-element contribution (Musaelian et al., 2023). Subsequently, we normalize the energy and force labels dataset-wise. Specifically, we divide the energy labels by their respective standard deviations, ensuring a standardized scale for comparison. The force labels are normalized by dividing them by the root-mean-square of the force components in the corresponding training set.

**Open Catalyst 2020:** The Open Catalyst 2020 (OC20) dataset (Chanussot et al., 2021) is a large and diverse catalyst dataset comprising a training set of 130 million examples, including 55 elements, 82 adsorbates, and catalysts consisting of unaries, binaries, and ternaries. The OC20 dataset consists of DFT relaxation trajectories, where the atom positions are iteratively updated based on the forces to minimize the energy. There are a total of 640k relaxations with an average trajectory length of 200, which makes the total training data $\sim$ 130M examples. In order to get better sampling across the potential energy surface, the dataset also contains ab initio molecular dynamics (AIMD) data and rattled data where atom positions are randomly perturbed. The cumulative size of all these data amounts to 189 million entries. All the density functional theory (DFT) calculations conducted in this study utilized VASP with $RPBE$ functional. Due to the considerable size of the pre-training dataset, we needed to establish practical training parameters. Consequently, we designated a training size of 100 million entries for our large training set, along with 2 million entries for our development set. These sizes were chosen to strike a balance between computational feasibility and the inclusion of a substantial amount of data for training purposes.

**Open Catalyst 2022:** The Open Catalyst 2022 (OC22) dataset, as presented in the work by Tran et al. (Tran et al., 2022), shares similarities with OC20 in terms of optimization trajectories involving adsorbates and catalyst surfaces. However, OC22 is more specialized and specifically focuses on oxide materials. While it may not possess the same level of diversity as OC20, it still provides valuable information within this specific context. The dataset comprises a total of 8M training data, which we employ in its entirety for our large run. Furthermore, we utilize 200k data from this dataset for our development run, ensuring a similar ratio of dataset sizes as our large run. It is important to note that for this study, a different DFT functional was employed compared to OC20. Specifically, the $PBE + U$ functional was utilized in this work. The decision to use this functional was made to account for the specific characteristics and properties of oxide catalysts, providing a more accurate representation of their behavior.

**ANI-1x:** The ANI-1x dataset (Smith et al., 2020) is a small molecule conformation dataset containing C, H, N, and O atoms, created using the Gaussian software with the $wb97x/6 - 31G(d)$ electronic structure method. This is a diverse organic dataset generated through active learning through a pre-training ML potential. This dataset has a total of 5M DFT calculations. We screened out organic molecules that have less than 4 atoms from the training data as our backbone GemNet model can't calculate quadruplets for these. We split the entire data into train, val, and test such that val and test have molecules that are not present in the train. We subsample 80k split for our development set out of the ~4M training data.

**Transition-1x:** The Transition 1x dataset (Schreiner et al., 2022) contains close to 10M DFT calculations of forces and energies of molecular configurations at the $wB97x/6 - 31G(d)$ level of theory (similar to ANI-1x). The configurations in this dataset are on and around reaction pathways generated by running Nudged Elastic Band (NEB) on 10k organic reactions. Therefore, this dataset contains a more dense sampling of PES for every system as compared to the ANI-1x dataset. The train, val, and test splits for this dataset are pre-defined and we use the same splits. For our development set, we subsample 200k split out of the total 9M train data.

## H.2 FINETUNING DATASETS

To demonstrate the ability of our pre-trained models to generalize over a diverse set of fine-tuning tasks, we selected two datasets from three different atomic domains. We include QMOF and Matbench from the materials domain, MD17 and QM9 from the small molecules domain, and SPICE (dipeptides and solvated amino acids subsets) and MD22 from the large molecules domain.

**QMOF:** The QMOF dataset (Rosen et al., 2021) is a database of approximately 15,000 experimentally synthesized metal organic frameworks (MOFs). We use a training dataset of 10,000 systems and split the remaining into validation and test sets. The band gap, which determines the electrical conductivity of a material, is an important property for identifying materials for electrocatalysis and energy applications, so we use it as the label for our models. It is worth noting that the non-referenced energy predictions for the other datasets (OC20, OC22, and ANI-1x) are extensive properties (i.e., the energy values depend on the size of the system), while the band gap is an intensive property (it does not depend on the size of the system). Therefore, we take a mean pooling of embeddings across nodes to calculate this scalar property.

**MatBench:** Matbench (Dunn et al., 2020) is a benchmark for predicting the properties of inorganic bulk materials. There are a total of 13 tasks in this benchmark that have samples that range in size from 312-132k samples. Tasks include predicting optical, thermal, electronic, thermodynamic, tensile and elastic properties given a material's composition and/or crystal structure. For our work, since we give structure as input to our model and demonstrate finetuning on regression tasks, we restrict to 8 tasks in Matbench. Each task has 5 folds and predefined test splits. We report an average across 5 folds for our JMP-L and JMP-S models but due to the compute cost, we stick to only reporting on fold 0 for all other comparisons. For prediction of all material properties across these tasks, we use mean pooling except for phonons. We observe max pooling to work better for phonons as the vibrational frequencies aren't intensive or extensive properties.

**MD17:** The MD17 (Chmiela et al., 2017) dataset, is a collection of eight small organic molecules for which energies and forces are computed using ab-initio Molecular Dynamics (MD) simulations with Density Functional Theory (DFT). The revised MD17 data is a recomputed version of the original MD17 with improved numerical accuracy. For this dataset, we use 950 samples for train, 50 samples for validation, and the remainder of the data as test set. This is the same number used by other models benchmarked on this dataset. All of our force predictions are modeled as the gradient of energies to achieve improved performances. Additionally, we also demonstrate results on a training split of 50 to demonstrate the few-shot learning capabilities of our pre-trained model (shown in Appendix B).

**QM9:** The QM9 dataset (Ramakrishnan et al., 2014) is a widely used benchmark dataset in the field of quantum chemistry and machine learning. It comprises a collection of quantum mechanical calculations for organic molecules containing up to nine heavy atoms from the GDB-17 database (Ruddigkeit et al., 2012b). The dataset provides essential molecular properties, including atomization energy, HOMO-LUMO gap, dipole moment, polarizability, and more. All molecules are modeled using the $B3LYP/6 - 31G(2df, p)$ DFT functional. We take an atomwise reference for all properties

and then normalize those to a standard Gaussian for predictions. We empirically find sum pooling to work better for all the property predictions.

**SPICE:** SPICE (Eastman et al., 2023) is a large and diverse dataset with the goal of training potentials relevant to simulating drug-like small molecules with proteins. It contains over a million conformers. We were interested in finetuning on the domain of larger molecules, we restricted our finetuning results to dipeptides and solvated amino acids subset. The dipeptides subset covers a full range of covalent interactions found in naturally occurring proteins and the solvated amino acids subset includes critical non-covalent interaction of protein-water and water-water.

**MD22:** MD22 (Chmiela et al., 2023) is a benchmark dataset for large molecules and includes molecules with sizes from 42 to 370 atoms. This dataset includes MD simulations of 8 molecules which include proteins, lipids, carbohydrates, nucleic acids, and supramolecules. All of these domains are not seen during pre-training. The trajectories for all the systems were sampled at temperatures between 400 and 500 K at a resolution of 1 fs, with corresponding potential energy and forces calculated at $PBE + MBD(61, 62)$ level of theory.

## H.3 DEVELOPMENT SPLITS FOR ABLATION STUDIES

Due to the high computational cost of training on our full pre-training and fine-tuning sets we made scaled-down development versions for our ablation studies. The pre-training subset includes a 2M split of OC20 as Gasteriger et al. (Gasteiger et al., 2022) demonstrate that results on this split correlate with the larger training split. Further, we chose splits of the other pre-training datasets that keep roughly the same ratio as present in the full pre-training set to enable the study of dataset imbalances. Additionally, for finetuning datasets, we pick a single target or molecule from each dataset. The development datasets are summarized in Table 17.

| Datasets | Task | Dev. train split |
|---|:---:|:---:|
| Pre-training datasets | | |
| OC20 | E, F | 2M |
| OC22 | E, F | 200k |
| ANI-1x | E, F | 80k |
| Transition-1x | E, F | 200k |
| Fine-tuning datasets | | |
| Matbench | MP E Form | 10k |
| QMOF | Band gap | 10k |
| MD17 | Aspirin Forces | 1k |
| QM9 | $\Delta\epsilon$ | 110k |
| SPICE | Solvated Amino Acids | 1k |
| MD22 | Stachyose | 8k |

Table 17: Scaled down development sets for ablation studies.

# I   DATASET OVERLAP

| Target (Units) | JMP-L Overlapping | JMP-L Non-overlapping | JMP-L Combined |
|---|---|---|---|
| $\mu$ $(D)$ | 0.008 | 0.006 | 0.008 |
| $\alpha$ $(a_0^3)$ | 0.032 | 0.030 | 0.032 |
| $\varepsilon_{\text{HOMO}}$ $(meV)$ | 9.1 | 7.7 | 8.8 |
| $\varepsilon_{\text{LUMO}}$ $(meV)$ | 8.5 | 8.9 | 8.6 |
| $\Delta\varepsilon$ $(meV)$ | 19.4 | 18.1 | 19.1 |
| $R^2$ $(a_0^2)$ | 0.162 | 0.164 | 0.163 |
| ZPVE $(meV)$ | 0.901 | 1.043 | 0.9 |
| $U_0$ $(meV)$ | 3.0 | 2.6 | 2.9 |
| $U$ $(meV)$ | 2.9 | 2.5 | 2.8 |
| $H$ $(meV)$ | 2.9 | 2.5 | 2.8 |
| $G$ $(meV)$ | 4.4 | 4.2 | 4.3 |
| $C_\nu$ (cal/mol K) | 0.017 | 0.018 | 0.017 |

Table 18: QM9 results where the test set is partitioned into overlapping and non-overlapping fractions based on composition.

As mentioned in the main text there is some overlap between the pre-training and fine-tuning small molecule datasets. This will likely always be the case for small molecules because under a certain number of heavy atoms nearly all possible molecules can be enumerated (Ruddigkeit et al., 2012a) and many datasets draw from this distribution. In particular, there is overlap between the molecules in ANI-1x and QM9. While there are some of the same or very similar molecules present, the level of DFT used to generate the data is different ($\omega$b97x vs B3LYP), the labels are not identical (although the thermodynamic properties are closely related to the total electronic energy), and the 3D structures may not be identical, given these differences it is unclear how the overlap will impact fine-tuning performance. To examine this further, we evaluated a pre-trained model that has been fine-tuned on QM9 (JMP-L) with multiple versions of test set, one with only the overlapping systems, one with all overlapping systems removed, and finally the combined or full test set. We consider the strictest case, where overlap is determined by composition i.e. if the same atoms are present. The results are hard to differentiate across all test splits, as shown in Table 18, indicating that model is learning not simply memorizing similar examples in pre-training.

# J   MODEL AND OPTIMIZATION HYPERPARAMETERS

Table 19 shows the model hyperparmeters for the small and large variants of the GemNet-OC backbone. Table 20 shows the training and optimization hyperparameters for the pre-training, fine-tuning, and scratch baseline training runs. Comma-separated hyperparameter values indicate that multiple values were evaluated in our experiments. In this case, an exhaustive grid search is conducted across all possible hyperparameters, and the hyperparameters that produce the best results (i.e., the lowest validation MAE scores) are selected. Notably, we use ReduceLROnPlateau with a patience of 3 and factor of 0.8 for all Scratch GN-OC runs and all GN-OC Small fine-tuning runs. For the FT GN-OC Large runs, we use the learning rate scheduling strategy as described in Appendix B.4. For rMD17 fine-tuning runs, we found that using a longer cosine decay duration of 128 epochs, with a lower cosine final LR factor of 1e-2, produces better results.

| GemNet-OC Hyperparameters | Small | Large |
|---|---|---|
| No. spherical basis | 7 | 7 |
| No. radial basis | 128 | 128 |
| No. blocks | 4 | 6 |
| Atom embedding size | 256 | 256 |
| Edge embedding size | 512 | 1024 |
| | | |
| Triplet edge embedding input size | 64 | 64 |
| Triplet edge embedding output size | 64 | 128 |
| Quadruplet edge embedding input size | 32 | 64 |
| Quadruplet edge embedding output size | 32 | 32 |
| Atom interaction embedding input size | 64 | 64 |
| Atom interaction embedding output size | 64 | 64 |
| Radial basis embedding size | 16 | 32 |
| Circular basis embedding size | 16 | 16 |
| Spherical basis embedding size | 32 | 64 |
| | | |
| No. residual blocks before skip connection | 2 | 2 |
| No. residual blocks after skip connection | 2 | 2 |
| No. residual blocks after concatenation | 1 | 4 |
| No. residual blocks in atom embedding blocks | 3 | 3 |
| No. atom embedding output layers | 3 | 3 |
| | | |
| Cutoff | 12.0 | 12.0 |
| Quadruplet cutoff | 12.0 | 12.0 |
| Atom edge interaction cutoff | 12.0 | 12.0 |
| Atom interaction cutoff | 12.0 | 12.0 |
| Max interaction neighbors | 30 | 30 |
| Max quadruplet interaction neighbors | 8 | 8 |
| Max atom edge interaction neighbors | 20 | 20 |
| Max atom interaction neighbors | 1000 | 1000 |
| | | |
| Radial basis function | Gaussian | Gaussian |
| Circular basis function | Spherical harmonics | Spherical harmonics |
| Spherical basis function | Legendre Outer | Legendre Outer |
| Quadruplet interaction | True | True |
| Atom edge interaction | True | True |
| Edge atom interaction | True | True |
| Atom interaction | True | True |

Table 19: Model hyperparameters for the small and large variants the GemNet-OC backbone model

Table 20: Optimization hyperparameters across different training runs.

| **Pre-Training JMP Small/Large** | |
|---|---|
| Batch Size | 1024/768 |
| Optimizer | AdamW |
| Weight Decay | 0.1 |
| EMA | 0.99 |
| Initial LR | 2.00e-4 |
| LR Scheduler | Warmup + Cos |
| Warmup Duration | 2000 steps |
| Warmup Starting Factor | 0.2 |
| Cos Duration | 2 epochs |
| Con Final LR Factor | 1.0e-1 |
| Max Training Epochs | 2 |

**Fine-Tuning JMP Small**

| | |
|---|---|
| Optimizer | AdamW |
| Weight Decay | 0.1 |
| EMA | 0.99 |
| Initial LR | 8.00e-5 |
| LR Scheduler | Warmup + Cos + LLRD + RLP |
| Warmup Duration | 5 epochs |
| Warmup Starting Factor | 1.00e-1 |
| Cos Duration | 32 epochs |
| Cos Annealing | false |
| Con Final LR Factor | 1.0e-1 |
| LLRD Embedding Block Initial LR Factor | 0.30 |
| LLRD Block 1 Initial LR Factor | 0.35 |
| LLRD Block 2 Initial LR Factor | 0.40 |
| LLRD Block 3 Initial LR Factor | 0.55 |
| LLRD Block 4 Initial LR Factor | 0.625 |
| RLP Patience | 5 |
| RLP Factor | 0.1 |

**Fine-Tuning JMP Large**

| | |
|---|---|
| Optimizer | AdamW |
| Weight Decay | 0.1 |
| EMA | 0.99 |
| Initial LR | 8.00e-5 |
| LR Scheduler | Warmup + Cos + LLRD + RLP |
| Warmup Duration | 5 epochs |
| Warmup Starting Factor | 1.00e-1 |
| Cos Duration | 32 epochs |
| Cos Annealing | false |
| Con Final LR Factor | 1.0e-1 |
| LLRD Embedding Block Initial LR Factor | 0.30 |
| LLRD Block 1 Initial LR Factor | 0.55 |
| LLRD Block 2 Initial LR Factor | 0.40 |
| LLRD Block 3 Initial LR Factor | 0.30 |
| LLRD Block 4 Initial LR Factor | 0.40 |
| LLRD Block 5 Initial LR Factor | 0.55 |
| LLRD Block 6 Initial LR Factor | 0.625 |
| RLP Patience | 5 |
| RLP Factor | 0.1 |

**Fine-Tuning JMP Large - MD17**

| | |
|---|---|
| Optimizer | AdamW |
| Weight Decay | 0.1 |
| EMA | 0.99 |
| Initial LR | 8.00e-5 |
| LR Scheduler | Warmup + Cos + LLRD + RLP |
| Warmup Duration | 5 epochs |
| Warmup Starting Factor | 1.00e-1 |
| Cos Duration | 32 epochs |
| Cos Annealing | false |
| Con Final LR Factor | 1.0e-2 |
| LLRD Embedding Block Initial LR Factor | 0.30 |
| LLRD Block 1 Initial LR Factor | 0.55 |
| LLRD Block 2 Initial LR Factor | 0.40 |
| LLRD Block 3 Initial LR Factor | 0.30 |
| LLRD Block 4 Initial LR Factor | 0.40 |
| LLRD Block 5 Initial LR Factor | 0.55 |
| LLRD Block 6 Initial LR Factor | 0.625 |
| RLP Patience | 5 |

| | |
|---|---|
| RLP Factor | 0.1 |

| **Scratch GN-OC Small and Large** | |
|---|---|
| Optimizer | AdamW |
| Weight Decay | 0.01 |
| EMA | 0.99 |
| Initial LR | 1e-4, 2e-4, 5e-5 |
| LR Scheduler | RLP |
| Patience | 3 |
| Factor | 0.8 |

