# OpenReview forum: "From Molecules to Materials: Pre-training Large Generalizable Models for Atomic Property Prediction"
_ICLR.cc/2024/Conference — ICLR 2024 poster_

### Official Review · Reviewer_avot · 2023-10-27

**Soundness:** 3 good
**Presentation:** 3 good
**Contribution:** 2 fair
**Rating:** 5
**Confidence:** 4

**Summary:**

The paper introduces a supervised pre-training strategy named JMP, which pretrains GemNet-OC on various small molecule data from multiple chemical domains. The pre-trained network acts as a foundational model, further finetuned for downstream atomic property prediction tasks. The authors’ major contribution is demonstrating that by pretraining exclusively on small molecules, the network can be finetuned on large molecule datasets, achieving state-of-the-art results. The primary technical advancements concentrate on refining each component of the standard deep learning pipeline, including data preprocessing and hyperparameter tuning. Experiments across 40 fine-tuning benchmarks are conducted to showcase the effectiveness of the method.

**Strengths:**

+ The paper is articulate and easily comprehensible. The authors provide an essential level of detail in describing their pipeline, facilitating a clear understanding of the processes. Although source code is not provided, the comprehensive details included in the paper should enable straightforward reimplementation.

+ The experiments presented in the paper are exhaustive and meticulous. A diverse array of molecular property datasets, encompassing both large and small molecules, has been utilized for the experiments. The authors have conducted extensive ablation studies, offering valuable insights into the influence of various hyperparameters used in the pipeline.

**Weaknesses:**

- My primary concern lies in the paper’s technical contribution. The concept of building a foundational model for molecular property prediction tasks isn’t novel. A significant challenge is bridging the substantial domain gaps across various chemistry domains. The authors seem to emphasize that the proposed pipeline can effectively bridge the molecule size gap, allowing it to work efficiently on larger molecules even when only pretrained on smaller ones. However, molecule size is just one of several apparent factors—and likely among the simpler ones—causing the domain gap. More complex factors, such as intrinsic differences in the distribution of graph structures and issues related to data availability, are not addressed in the paper. Consequently, it is challenging to be convinced that the proposed method significantly contributes to foundational models or represents "an important step for universal ML potential," as claimed in the introduction.

- The improvement brought by the proposed method appears to be mainly attributed to hyperparameter tuning. The network architecture itself isn’t novel, and the loss function closely resembles commonly used ones, albeit with slight modifications to some coefficients. My overarching impression is that the authors engage extensively in manual hyperparameter tuning, which doesn’t offer substantial insights to propel further research advancements. While I acknowledge the empirical enhancements demonstrated through comprehensive experiments, it is still challenging to bestow a favorable overall evaluation on the paper.

**Questions:**

1. In Sect. 4.1, concerning Data Normalization, the authors have chosen to normalize the property values per dataset. A lingering question is how the output of the NN is transformed. Is the transformation still dependent on each specific dataset? If that is the case, it seems impractical for real world applications where a novel molecule is given, and it would be indeterminable as to which "dataset" it inherently belongs to and how to transform its output.

2. Regarding Dataset Size Imbalance, I was wondering if the authors considered utilizing loss reweighting as opposed to data reweighting. By loss reweighting, I am referring to the approach of uniformly sampling the data but adjusting the coefficients of each sample to p_d (ensuring normalization across each batch).

3. I devoted a significant amount of time attempting to digest whether each term in Eq.1 is a novel contribution or a previously introduced one. It would be beneficial if the authors could provide clearer definitions of each symbol used, elaborate more distinctly on the novel improvements introduced in this paper, and add a period to the end of the equation.

---

> ### Author Response · Authors · 2023-11-17
> **Response to Reviewer avot: 1/4**
>
> We thank the reviewer for their time and effort in reviewing our paper. We have addressed the reviewer's comments below.
>
> > My primary concern lies in the paper's technical contribution. The concept of building a foundational model for molecular property prediction tasks isn't novel.  [...] Consequently, it is challenging to be convinced that the proposed method significantly contributes to foundational models or represents "an important step for universal ML potential," as claimed in the introduction.
>
> We refer the reviewer to our response to the general response for an in-depth discussion on novelty. Our model is pre-trained on diverse data from the small molecule and catalysis domains. This diversity presents major challenges in pre-training, and we address these challenges through our proposed pre-training pipeline. This formulation is what enables our model to generalize across a diverse set of downstream tasks and chemical domains. We believe that this is a significant contribution to the community.
>
> > A significant challenge is bridging the substantial domain gaps across various chemistry domains. The authors seem to emphasize that the proposed pipeline can effectively bridge the molecule size gap, allowing it to work efficiently on larger molecules even when only pretrained on smaller ones. However, molecule size is just one of several apparent factors—and likely among the simpler ones—causing the domain gap. More complex factors, such as intrinsic differences in the distribution of graph structures and issues related to data availability, are not addressed in the paper.
>
> We are unsure what the reviewer means by "intrinsic differences in the distribution of graph structures". Our pre-training and fine-tuning datasets contain extremely diverse graph structures, including small molecules, large molecules, materials, and catalyst systems. This diversity covers much more than just the molecule size gap. Please see Table 1 for more detailed information on the datasets and Figure 1 (left and middle) for visualizations of sample systems from each dataset. As can be seen, these systems are extremely diverse in terms of atom count, atom types, graph structure, and chemical domain. If the reviewer believes that there are other factors that we have not considered, we would be happy to discuss further.
>
> Regarding the issue of data availability, our state-of-the-art results on low-resource datasets (e.g., rMD17, MatBench's JDFT2D, etc.) demonstrate that the pre-trained model is able to generalize to low-resource datasets, decreasing the need for large amounts of data. We believe that this is a significant contribution to the community.

---

> > ### Author Response · Authors · 2023-11-17
> > **Response to Reviewer avot: 2/4**
> >
> > > The improvement brought by the proposed method appears to be mainly attributed to hyperparameter tuning. [...] My overarching impression is that the authors engage extensively in manual hyperparameter tuning, which doesn’t offer substantial insights to propel further research advancements. While I acknowledge the empirical enhancements demonstrated through comprehensive experiments, it is still challenging to bestow a favorable overall evaluation on the paper.
> >
> > We strongly disagree with this statement. For pre-training, our model uses the nearly identical hyperparameters to the original GemNet-OC large model, with the only major difference being that we use a cosine learning rate schedule, instead of GemNet-OC's ReduceLROnPlateau schedule. This is because the traditional ReduceLROnPlateau schedule is not well-defined in the multi-task pre-training setting, where there is no single validation metric to monitor.
> > For fine-tuning, due to the extremely large number of downstream tasks (40 in total, with the 8 MatBench tasks having 5 folds each, yielding 72 total downstream tasks), we were unable to perform extensive hyperparameter tuning for each task. Instead, we performed a single hyperparameter sweep on the QMOF dataset and repeated the same hyperparameters for all other tasks, with the sole exception being the rMD17 dataset, for which we use a more conservative ReduceLROnPlateau patience and stopping criteria taken from the previous SOTA work, Allegro [Musaelian et al., 2022](https://arxiv.org/abs/2204.05249). It is very likely that the hyperparameters we used for the other tasks are not optimal for those tasks and that further hyperparameter tuning would yield better results.
> >
> > We juxtapose this with other SOTA methods that we're comparing our results to:
> > - Nearly all existing SOTA models (e.g., [Gasteiger et al., 2022](https://arxiv.org/abs/2106.08903), [Gasteiger et al., 2022 (b)](https://arxiv.org/abs/2204.02782), [Musaelian et al., 2022](https://arxiv.org/abs/2204.05249), [Thölke et al., 2022](https://arxiv.org/abs/2202.02541), [Liao et al., 2023](https://arxiv.org/abs/2206.11990)) use different hyperparameters (e.g., learning rate and LR schedules, number of layers, underlying distance encoding (e.g., RBF) function and number of RBF kernels, radius graph cutoff, feature dimension, etc.) when evaluating different datasets.
> > - Many go even further, making major changes to hyperparameters for different tasks *within the same dataset*. For example, [Liao et al., 2023](https://arxiv.org/abs/2206.11990) uses a different learning rate, dropout ratio, weight decay, batch size, and stopping criteria for different tasks within the same dataset (see Appendix D and E of their paper for QM9 and MD17, respectively).
> >
> > We believe that our approach to selecting hyperparameter is principled and reproducible. Our results are not cherry-picked to maximize performance on a particular dataset or task. We use the same hyperparameters for all datasets and tasks, and we do not perform any independent hyperparameter tuning for each downstream task. We believe that this is a more fair comparison, and it is more likely to yield generalizable results. The impressive results of our model on the PDBBind dataset of MoleculeNet --- a suggestion from reviewer `XFKt` --- further demonstrates the generalizability of our model.

---

> > > ### Author Response · Authors · 2023-11-17
> > > **Response to Reviewer avot: 3/4**
> > >
> > > > The network architecture itself isn’t novel.
> > >
> > > The goal of this paper is not to introduce a novel network architecture. Instead, we aim to demonstrate the effectiveness of diverse multi-task pre-training on downstream tasks. Please see our response to reviewer `XFKt` for our reasoning behind using the GemNet-OC model.
> > >
> > > > In Sect. 4.1, concerning Data Normalization, the authors have chosen to normalize the property values per dataset. A lingering question is how the output of the NN is transformed. Is the transformation still dependent on each specific dataset? If that is the case, it seems impractical for real world applications where a novel molecule is given, and it would be indeterminable as to which "dataset" it inherently belongs to and how to transform its output.
> > >
> > > Our goal for this work is to build a strong backbone model through pre-training that has learned molecular and materials representations. Similar to how BERT's pre-trained masked language modeling head is discarded and replaced with a new task-specific head, we discard the pre-trained regression head and replace it with a new task-specific head.
> > >
> > > > Regarding Dataset Size Imbalance, I was wondering if the authors considered utilizing loss reweighting as opposed to data reweighting. By loss reweighting, I am referring to the approach of uniformly sampling the data but adjusting the coefficients of each sample to p_d (ensuring normalization across each batch).
> > >
> > > Regarding dataset size imbalance, we initially considered loss re-weighting. Ultimately, we decided to use data re-weighting for a couple reasons:
> > >
> > > 1. Data re-weighting is a simple and common technique used in multitask learning to handle dataset size imbalance. Many prominent multilingual NLP works use data re-weighting via temperature sampling to handle imbalanced corpora across languages. Since our problem setup is similar (imbalanced datasets across chemical tasks), we opted for the tried and tested technique of data re-weighting.
> > > 2. In early experiments, we found that loss re-weighting can be tricky to tune properly. The loss coefficients need to be calibrated correctly to handle the size imbalance, and simply using $p_d$ as the loss coefficient does not yield good results. We found data re-weighting to be more stable and effective than loss re-weighting for our setup.

---

> > > > ### Author Response · Authors · 2023-11-17
> > > > **Response to Reviewer avot: 4/4**
> > > >
> > > > > I devoted a significant amount of time attempting to digest whether each term in Eq.1 is a novel contribution or a previously introduced one. It would be beneficial if the authors could provide clearer definitions of each symbol used, elaborate more distinctly on the novel improvements introduced in this paper, and add a period to the end of the equation.
> > > >
> > > > All the relevant notation used in Eq. 1 is defined in the text: Either in the "Notation" section or in the corresponding loss coefficient subsections. The proposed change is very subtle but extremely important. Below, we have added sample Python code for clearly describing our proposed loss function. The table below maps the notation used in the code to the notation used in the paper.
> > > >
> > > > | **Sample Code Notation** | **Paper Notation**           |
> > > > | ------------------------ | ---------------------------- |
> > > > | `E`                      | $E$                          |
> > > > | `F`                      | $F$                          |
> > > > | `E_target`               | $\hat{E}$                    |
> > > > | `F_target`               | $\hat{F}$                    |
> > > > | `lambda_E`               | $\lambda_E$                  |
> > > > | `lambda_F`               | $\lambda_F$                  |
> > > > | `batch_idx`              | Not present (see note below) |
> > > > | `dataset_idx`            | $W$                          |
> > > >
> > > > The `batch_idx` tensor is a product of PyTorch Geometric's sparse graph batching, which constructs a batched graph as one large graph with disconnected components. The `batch_idx` tensor maps each node in the batched graph to its corresponding component in the original batch. In the notation of the paper, we do not use this explicitly to simplify the notation. Instead, we use subscripts on node-level quantities to denote the component that the node belongs to. For example, $F_{b,i}$ denotes the force vector of the $i$-th atom in the $b$-th component of the batched graph, or $N_{b}$ denotes the number of atoms in the $b$-th component of the batched graph.
> > > >
> > > > ```python
> > > > def gemnet_oc_loss(
> > > >     E: Tensor, # (batch_size,)
> > > >     F: Tensor, # ((batch_size*num_atoms), 3)
> > > >     E_target: Tensor, # (batch_size,)
> > > >     F_target: Tensor, # ((batch_size*num_atoms), 3)
> > > >     batch_idx: Tensor, # ((batch_size*num_atoms),)
> > > >     lambda_E: Tensor, # scalar
> > > >     lambda_F: Tensor, # scalar
> > > > ):
> > > >     # Energy loss computation
> > > >     systemwise_energy_loss = (E - E_target) ** 2 # (batch_size,)
> > > >     energy_loss = lambda_E * torch.mean(
> > > >         systemwise_energy_loss
> > > >     ) # scalar
> > > >
> > > >     # Force loss computation
> > > >     atomwise_F_loss = torch.norm(
> > > >         F - F_target,
> > > >         p=2,
> > > >         dim=-1
> > > >     ) # (batch_size*num_atoms,)
> > > >     force_loss = lambda_F * torch.mean(atomwise_F_loss) # scalar
> > > >
> > > >     return energy_loss + force_loss
> > > >
> > > > def jmp_loss(
> > > >     E: Tensor, # (batch_size,)
> > > >     F: Tensor, # ((batch_size*num_atoms), 3)
> > > >     E_target: Tensor, # (batch_size,)
> > > >     F_target: Tensor, # ((batch_size*num_atoms), 3)
> > > >     batch_idx: Tensor, # ((batch_size*num_atoms),)
> > > >     dataset_idx: Tensor, # (batch_size,)
> > > >     lambda_E: Tensor, # (num_datasets,)
> > > >     lambda_F: Tensor, # (num_datasets,)
> > > > ):
> > > >     # Energy loss computation
> > > >     # Energy computation is the same as the
> > > >     #   original GemNet-OC loss function,
> > > >     #   with the only difference being that
> > > >     #   we have task-specific energy loss coefficients.
> > > >     systemwise_energy_loss = (E - E_target) ** 2 # (batch_size,)
> > > >     energy_loss = torch.mean(
> > > >         systemwise_energy_loss
> > > >         * lambda_E[dataset_idx]
> > > >     ) # scalar
> > > >
> > > >     # Force loss computation
> > > >     atomwise_F_loss = torch.norm(
> > > >         F - F_target,
> > > >         p=2,
> > > >         dim=-1
> > > >     ) # (batch_size*num_atoms,)
> > > >     # The difference is that we now
> > > >     #   perform a structure-level averaging,
> > > >     #   instead of an atom-level averaging.
> > > >     structurewise_force_loss = scatter_mean(
> > > >         atomwise_F_loss,
> > > >         batch_idx
> > > >     ) # (batch_size,)
> > > >     # Similarly to the energy loss, we now have
> > > >     #   task-specific force loss coefficients.
> > > >     force_loss = torch.mean(
> > > >         structurewise_force_loss
> > > >         * lambda_F[dataset_idx]
> > > >     ) # scalar
> > > >
> > > >     return energy_loss + force_loss
> > > > ```
> > > >
> > > > We have updated our Appendix to include this sample code and a more detailed description of our loss function.

---

> > > > > ### Comment · Reviewer_avot · 2023-11-22
> > > > >
> > > > > Thank you for the authors' rebuttal. I appreciate the thorough clarification and the addition of new experimental results. My apologies for not responding sooner. I now recognize that the method is not as reliant on hyperparameter tuning as I initially thought, and I acknowledge its practical contribution in providing a model with decent performance. So I have revised my score from 3 to 5.
> > > > >
> > > > > However, I maintain a slightly negative stance due to my concerns about the novelty of the paper. Despite understanding the authors' general response and the discussions with other reviewers, the technical novelty seems somewhat limited. The use of reweighting and data normalization appears to be more aligned with engineering tricks rather than substantial technical innovations that can offer deep insights or potential inspiration for future research. While assessing novelty can be subjective, I believe that for a paper to qualify for ICLR, the novelty of the method should be a significant consideration, surpassing mere engineering or experimental performance.

---

> > > > > > ### Author Response · Authors · 2023-11-23
> > > > > >
> > > > > > We appreciate reviewer `avot` revising their score and recognizing the practical contribution of our work. We believe our approach does contribute technical innovations beyond “engineering tricks”, as it offers a viable blueprint for developing foundation models pre-trained across chemical domains — something that did not previously exist — and opens up a new area of research. Additionally, we provide excellent empirical results and thorough benchmarking to gain a better understanding of generalization.
> > > > > >
> > > > > > While novelty can be subjective, we believe our work clearly crosses the threshold. There are numerous examples from top venues like ICLR where ideas were taken from one field and applied to another e.g. NLP to CV [[Dosovitskiy et al., ICLR 2021](https://openreview.net/forum?id=YicbFdNTTy), [Bao et al, ICLR 2022](https://openreview.net/forum?id=p-BhZSz59o4)]. In these instances, the technical contributions may be conceptually straightforward, but they offer significant benefits in practice and open up new research directions. One additional point of reference within the area of pre-training for atomic systems is the work by [Zaidi et al., 2023](https://openreview.net/forum?id=tYIMtogyee) --- accepted at ICLR 2023 with very positive reviews (notable-top-25%) --- that builds on prior work and uses the same (or very similar) underlying architecture, denoising technique, and loss function [[Godwin et al., 2022](https://arxiv.org/abs/2106.07971)]. Despite the straightforward nature of these approaches, they have turned out to be very influential in their respective communities. We envision our work as having the potential to be similarly impactful and making a valuable contribution to the machine learning for chemistry community.

---

### Official Review · Reviewer_R7em · 2023-10-31

**Soundness:** 3 good
**Presentation:** 4 excellent
**Contribution:** 3 good
**Rating:** 8
**Confidence:** 4

**Summary:**

Authors explore the application of machine learning in predicting atomic properties across a wide array of applications, from healthcare to climate change. The authors introduce Joint Multi-domain Pre-training (JMP), a supervised pre-training strategy that leverages a vast dataset comprising approximately 120 million examples from multiple chemical domains. The primary goal of JMP is to generate transferable atomic representations that can be fine-tuned for diverse downstream tasks, addressing the challenge of generalizing across the extensive and complex space of molecular interactions.

**Strengths:**

Innovative Approach: The paper introduces Joint Multi-domain Pre-training (JMP), a novel supervised pre-training strategy that leverages a massive dataset from multiple chemical domains. This approach is innovative in its attempt to generate transferable atomic representations for a wide array of downstream tasks, addressing the challenge of generalizing across diverse molecular interactions.

Creative Combination of Ideas: The authors draw inspiration from successful practices in Natural Language Processing (NLP) and Computer Vision (CV), creatively applying the concept of large-scale pre-training to the domain of atomic property prediction. This cross-disciplinary innovation enhances the originality of the work.

Broad Applicability: The paper’s contributions have broad applicability across various domains, ranging from drug discovery to material science. The ability of JMP to generalize across diverse chemical domains signifies its potential to drive advancements in multiple fields.
Addressing a Critical Challenge: The paper tackles the critical challenge of generating transferable atomic representations in the vast and complex space of molecular interactions. By addressing this challenge, the paper makes a significant contribution to the field of machine learning for atomic modeling.

Computational Efficiency: The computational efficiency achieved through JMP, with over 12x faster fine-tuning compared to training from scratch, is a notable strength. This efficiency is crucial for practical applications, making the paper’s contributions highly significant.

**Weaknesses:**

Need for Broader Ablation Studies:
Issue: While the paper includes ablation studies to analyze the impact of different JMP components, these studies could be broadened to provide a more comprehensive understanding of the model’s behavior and the contributions of individual components.

**Questions:**

N/A

---

> ### Author Response · Authors · 2023-11-17
> **Response to Reviewer R7em: 1/1**
>
> We thank the reviewer for their time and effort in reviewing our paper. Specifically, we are glad that the reviewer found our work to be innovative, creative, and broadly applicable.
>
> > Need for Broader Ablation Studies: Issue: While the paper includes ablation studies to analyze the impact of different JMP components, these studies could be broadened to provide a more comprehensive understanding of the model’s behavior and the contributions of individual components.
>
> In our current manuscript (see section 5.1), we have conducted the necessary ablations to analyze the impacts of the core components of our pre-training technique. We also refer the reviewer to Appendix B, where we have conducted a number of additional ablations.
>
> With that said, additional experiments, such as a "scaling law" analysis of model size and pre-training dataset size, would be very interesting and informative. However, due to the high computational cost of our experiments, we were unable to perform these additional experiments. In particular, our JMP-L model took 34400 GPU hours to train, and the fine-tuning experiments took 4600 GPU hours (for all 72 fine-tuning datasets). We have updated the conclusion in our manuscript to reflect this limitation and have added a note in the conclusion section that a more comprehensive ablation study is left for future work.
>
> Regarding model-architecture-level analysis, we heavily leverage the findings of the original GemNet and GemNet-OC works (see [Gasteiger et al., 2022](https://arxiv.org/abs/2106.08903) and [Gasteiger et al., 2022 (b)](https://arxiv.org/abs/2204.02782)), which do a very thorough job of exploring the impact of different components of the underlying model GemNet architecture across different datasets. See our response to reviewer `XFKt` for further discussion on our backbone architecture choice.

---

> > ### Comment · Reviewer_R7em · 2023-11-21
> >
> > Thanks for the work. I have no further questions.

---

### Official Review · Reviewer_KadD · 2023-11-02

**Soundness:** 3 good
**Presentation:** 2 fair
**Contribution:** 2 fair
**Rating:** 5
**Confidence:** 4

**Summary:**

This paper pretrains a large generalizable model for atomic property prediction, which outperforms SOTA methods on many downstream tasks.

**Strengths:**

1. The model achieves exceptional performance.
2. The experiments conducted in this study are comprehensive and thorough, covering various aspects such as hyper-parameter settings, ablation studies, and downstream tasks.
3. The authors carefully study the balance between different datasets and pre-training tasks, ensuring a comprehensive analysis of their impact.

**Weaknesses:**

1. The disscussion of the correlation between pre-training tasks and downstream tasks is missing. I am wondering what kinds of downstream tasks can be promoting by pre-training?
2. The main contribution of this work is implemental and not suprising.
3. This paper confuses the prediction of atomic properties with the prediction of molecular properties.

**Questions:**

1. Does the use of JMP contribute to the prediction of molecular properties? If so, what specific types of molecular properties show improvements?

---

> ### Author Response · Authors · 2023-11-17
> **Response to Reviewer KadD: 1/1**
>
> We thank the reviewer for their positive comments, and we are glad of the recognition of our model's "exceptional performance" and that they found our work to be "comprehensive and thorough". We address each of the reviewer's comments below.
>
> > The disscussion of the correlation between pre-training tasks and downstream tasks is missing.
>
> In Section 3 (Datasets), we discuss the differences between the pre-training and fine-tuning datasets in two different aspects:
> 1. We identify the underlying chemical domain of the pre-training and fine-tuning datasets and show that our fine-tuning datasets span a wide range of chemical domains, including ones that are out-of-domain for the pre-training datasets.
> 2. Our pre-training datasets all contain energy and force labels for force-field modeling. Our fine-tuning datasets, on the other hand, contain a diverse mix of labels, including atomization energy, polarizability, electron affinity, formation energy, and band gap, in addition to energy and force labels (for the DFT and MD datasets). We refer to these other labels as "out-of-domain labels" in the manuscript.
>
> We have also updated the introduction of our manuscript to further clarify the differences amongst the chemical domains that we pre-train and fine-tune our model on.
>
> > I am wondering what kinds of downstream tasks can be promoting by pre-training?
>
> As we show in Figure 2 (a), when compared to training from scratch, pre-training helps with all of the downstream tasks we consider, including all in-domain (ID) and out-of-domain (OOD) labels, as well as all ID and OOD chemical domains. However, if there are specific tasks you are interested in or if we have misunderstood your question, could you please provide more details or clarify? We would be happy to provide a more targeted response.
>
>
> > The main contribution of this work is implemental and not suprising.
>
> We refer the reviewer to our response to the general response for further discussion on novelty. On the results being "not surprising": We believe that our works makes a significant contribution to the community, but we are happy to get more insights on additional experiments required to help demonstrate the efficacy of this approach.
>
> > This paper confuses the prediction of atomic properties with the prediction of molecular properties.
>
> If the reviewer is referring to the difference between atom-level (node-level) and system-level\*\*\* (graph-level) properties, we would like to point out that our set of downstream tasks cover atomic properties (e.g., atom-level forces) and molecular properties (e.g., dipole moment in QM9). Our pre-training objective also involves atomic properties (forces) and molecular properties (energy). Otherwise, we would like to ask the reviewer to further clarify their comment regarding the confusion between atomic and molecular properties.
>
> \*\*\* We use the term "system" or "graph" as opposed to "molecule" because some of our datasets are not just molecules, but also materials (e.g., MatBench), catalyst systems (e.g., OC20), among others. We have clarified this in the manuscript.
>
> > Does the use of JMP contribute to the prediction of molecular properties? If so, what specific types of molecular properties show improvements?
>
> Please see our response to the previous comment regarding molecular vs. atomic properties. Regarding observed improvements, as mentioned previously (and as shown in Figure 2 (a)), pre-training helps with all of the downstream tasks we consider compared to training from scratch, including atom-level and molecule-level properties.

---

### Official Review · Reviewer_XFKt · 2023-11-04

**Soundness:** 3 good
**Presentation:** 3 good
**Contribution:** 3 good
**Rating:** 5
**Confidence:** 4

**Summary:**

This paper explores the possibility of pre-training a foundation-style model over multiple chemical domains to generate transferable atomic representations for downstream fine-tuning tasks. The Joint Multi-domain Pre-training (JMP) strategy utilizes data from multiple chemical domains and achieves state-of-the-art results across many targets of various datasets. The paper establishes a comprehensive set of fine-tuning benchmarks across various chemical domains and tasks.

**Strengths:**

- Comprehensive set of fine-tuning benchmarks: The paper establishes a comprehensive set of fine-tuning benchmarks across various chemical domains and tasks, which enables researchers to evaluate the performance of their models against a standardized set of benchmarks.

- State-of-the-art results: The Joint Multi-domain Pre-training (JMP) strategy achieves state-of-the-art results across many targets of various datasets, which demonstrates the effectiveness of the proposed approach.

- Large and diverse molecular datasets: The paper highlights the importance of large and diverse molecular datasets in enabling the development of accurate and efficient models for atomic property prediction.

- Different model sizes: The paper provides multiple model sizes with pretrained checkpoints that can benefit real-world deployment at different resource levels and potentially accelerate research progress in related fields.

**Weaknesses:**

- The paper claims that the proposed pre-training method is model-agnostic. However, the only evaluated architecture backbone is GemNet-OC. It would be better to have a variant pre-trained using other types of model backbones to conduct further comparison and analysis.

- The novelty is a bit limited. I admit that this paper contributes on providing the empirical evidence that pre-training cross-domain molecule data can benefit multiple downstream tasks. However, the techniques used in this paper are either introduced by other literature or very simple and straightforward. No novel methods/theories are introduced. I would recommend this paper submit to more domain-related or comprehensive journals.

- This paper claims that many previous efforts on pretraining molecules focus on a specific domain which ignores the information provided by other domains and generalizability. It would be better to provide more empirical evidence that compares the proposed model with other pre-training methods [1].

- More supervised SOTA should be compared. E.g., for the materials domain, there are two more recent papers [2, 3] that can be reported against the proposed method.

- I understand the concerns of releasing the code and models before acceptance. However, in terms of reproducibility, it would be better to provide an anonymized repo including some demo models for testing purpose.

[1] Xia, Jun, et al. "A Systematic Survey of Chemical Pre-trained Models." IJCAI, 2023.

[2] Yan, Keqiang, et al. "Periodic graph transformers for crystal material property prediction." Advances in Neural Information Processing Systems 35 (2022): 15066-15080.

[3] Lin, Yuchao, et al. "Efficient Approximations of Complete Interatomic Potentials for Crystal Property Prediction." arXiv preprint arXiv:2306.10045 (2023).

**Questions:**

- Can you provide more details about the scaling strategy of the model architecture? The message-passing paradigm suffers from over-smoothing a lot and it is notorious of hard to make the network deep. I would like to understand more about this and how this method overcome the issues.

- Since for larger foundation models, we would like to include far more parameter. But the molecular graphs often just include a very small number of vocabularies. A better strategy could be adding full attention mechanism to make the parameter space larger. And the model would be a transformer-based architecture or so-called "graph transformer". Did the authors try this in their experiments? How did the two frameworks perform?

- This paper claims that they can deal with OOD challenge and cross-domain adaption better and they can benefit drug discovery, etc. So I would like to see more results on how this method perform on other therapeutics data [4], e.g., the molecule property predictions (toxicity, solubility, lipophilicity, etc.) on MoleculeNet [5].

[4] Huang, K., Fu, T., Gao, W. et al. Artificial intelligence foundation for therapeutic science. Nat Chem Biol 18, 1033–1036 (2022). https://doi.org/10.1038/s41589-022-01131-2

[5] Wu, Zhenqin, et al. "MoleculeNet: a benchmark for molecular machine learning." Chemical science 9.2 (2018): 513-530.

---

> ### Author Response · Authors · 2023-11-17
> **Response to Reviewer XFKt: 1/5**
>
> We thank the reviewer for their thoughtful feedback and highlighting our comprehensive benchmarks with state-of-the-art results, our choice of large and diverse pre-training dataset, and our experiments with model sizes. We also appreciate the questions and concerns raised which we address below.
>
> > The paper claims that the proposed pre-training method is model-agnostic. However, the only evaluated architecture backbone is GemNet-OC. It would be better to have a variant pre-trained using other types of model backbones to conduct further comparison and analysis.
>
> The primary goal of this work was to explore whether or not it is possible to learn robust and generalizable representations that work across many domains, which we were able to demonstrate for GemNet. When this work began we chose GemNet as the base architecture for a number of reasons including: (1) its application (or that of its predecessor, DimeNet) and performance across a diverse set of chemical domains and  downstream tasks, such as on OC20, OC22, QM9, MD17, MatBench, PDBBind (2) its performance on large diverse datasets such as OC20 and OC22 [[Gasteiger et al., 2022](https://arxiv.org/abs/2106.08903), [Tran et al., 2023](https://arxiv.org/abs/2206.08917)] (3) its ability to improve with a large number of parameters [[Sriram et al., 2022]](https://arxiv.org/abs/2203.09697), and (4) it is a reasonably efficient model in terms of training time. Even with (4) pre-training our largest model required 34400 GPU hours and 4600 GPU hours for fine-tuning models across all 72 fine-tuning datasets (which contains 40 unique tasks, and MatBench tasks having 5 folds). Given these compute numbers it was not feasible for us to include another model.
>
> > The novelty is a bit limited. I admit that this paper contributes on providing the empirical evidence that pre-training cross-domain molecule data can benefit multiple downstream tasks. However, the techniques used in this paper are either introduced by other literature or very simple and straightforward. No novel methods/theories are introduced.
>
> We are the first to demonstrate that representations can indeed generalize when pre-trained across a wide range of atomic domains. These domains vary from molecules to materials, have a large variance in the number of atoms, and even use different levels of theory to perform their calculations. The novelty lies in framing a supervised pre-training framework for these diverse datasets and is bolstered by comprehensive benchmarks that leads a way to future work in this field. We refer the reviewer to our response to the general response for further discussion on novelty.

---

> > ### Author Response · Authors · 2023-11-17
> > **Response to Reviewer XFKt: 2/5**
> >
> > > This paper claims that many previous efforts on pretraining molecules focus on a specific domain which ignores the information provided by other domains and generalizability. It would be better to provide more empirical evidence that compares the proposed model with other pre-training methods
> >
> > We agree that making comparisons with other pre-training methods is important. We have tried to do that wherever results were reported. In the main paper, we compare with two other pre-training methods in both the QM9 and the QMOF tables. Additionally, Figure 3 compares the effectiveness of pre-training methods to improve performance in relation to their trained from scratch counterpart for two QM9 targets. Given that we found limited points of comparison in the literature, we also fine-tuned the pre-trained Equivariant Transformer checkpoint (labeled as Pre-trained ET+NN) from [Zaidi et al., 2023](https://openreview.net/forum?id=tYIMtogyee) on our set of downstream tasks. We reported results on rMD17, QM9, MD22, and SPICE in Appendix A.
> > We have also fine-tuned on QMOF and MatBench but, out of fairness to the authors, we have excluded these results from our original manuscript because [Zaidi et al., 2023](https://openreview.net/forum?id=tYIMtogyee)'s pre-training dataset is the PCQM4Mv2 dataset, which consists of ~3M molecules, whereas QMOF and MatBench contain completely out-of-domain materials data. These results are included below for the reviewer's reference. The missing QMOF results for Pre-trained ET+NN are due to runs failing or hitting NaNs, despite our repeated attempts to re-run the experiments.
> >
> > | Materials (Units)            | Pretrained ET+NN (fold0 / mean) | JMP-S (fold0 / mean) | JMP-L (fold0 / mean) |
> > | ---------------------------- | ------------------------------- | -------------------- | -------------------- |
> > | JDFT2D ($meV/atom$)          | 37.62 / 55.37                   | 20.72 / 30.16        | 23.12 / **29.94**    |
> > | Phonons ($cm^-1$)            | 229.46 / 203.88                 | 26.6 / 22.77         | 21.28 / **20.57**    |
> > | Dielectric ($unitless$)      | 0.482 / 0.582                   | 0.133 / 0.252        | 0.119 / **0.249**    |
> > | Log GVRH ($log10(GPA)$)      | 0.130 / 0.138                   | 0.060 / 0.062        | 0.057 / **0.059**    |
> > | Log KVRH ($log10(GPA)$)      | 0.112 / 0.115                   | 0.044 / 0.046        | 0.045 / **0.045**    |
> > | Perovskites ($eV/unit cell$) | 0.541 / 0.542                   | 0.029 / 0.028        | 0.026 / **0.026**    |
> > | MP Gap ($eV$)                | 0.573 / 0.580                   | 0.119 / 0.121        | 0.089 / **0.091**    |
> > | MP E Form ($meV/atom$)       | 44.2 / 42.4                     | 13.6 / 13.3          | 10.3 / **10.1**      |
> > |                              |                                 |                      |                      |
> > | QMOF Band Gap                | -                               | 0.180                | **0.160**            |

---

> > > ### Author Response · Authors · 2023-11-17
> > > **Response to Reviewer XFKt: 3/5**
> > >
> > > > This paper claims that they can deal with OOD challenge and cross-domain adaption better and they can benefit drug discovery, etc. So I would like to see more results on how this method perform on other therapeutics data [4], e.g., the molecule property predictions (toxicity, solubility, lipophilicity, etc.) on MoleculeNet [5]
> > >
> > > Our current results  demonstrate that our model can generalize across multiple domains: We pre-train on the catalysis and small molecule domains and fine-tune on small molecules (ID), large molecules (OOD), and materials (OOD).
> > >
> > > Regarding drug discovery, the QM9 dataset [[Ramakrishnan et al., 2014]](https://www.nature.com/articles/sdata201422), for which we show SOTA performance on 11/12 tasks, is a dataset of small drug-like molecules, created for the purpose of de-novo drug discovery. More generally, force-field modeling and molecular dynamics simulations are important tasks in drug discovery, and we show SOTA performance on these tasks as well (see our MD17, MD22, and SPICE results).
> > >
> > > Regarding MoleculeNet, we agree that molecule property predictions (e.g., toxicity, solubility, lipophilicity) are important tasks in drug discovery. However, most of the datasets within MoleculeNet do not provide 3D structures (i.e., only SMILES strings are provided). Our pre-training strategy and underlying model was designed to operate on 3D structures. While it is possible to generate 3D structures from SMILES strings, this is a two-step process, and the model's performance will heavily rely on the quality of the 3D structure generation. We did not want to add this to our work as (1) it would be difficult to differentiate the true impact of this work with prior work and (2) it would be out of scope for this paper. The only datasets in MoleculeNet that provide 3D structures are QM7, QM8, QM9, and PDBBind (binding affinity).  We have already included QM9 and the molecules in QM7 and QM8 are quite similar. Using molecular compositions (i.e., atom counts) to identify overlaps, we found 89.5% of QM7’s molecules and 100% of QM8’s molecules overlap with those in QM9.
> > > For PDBBind, we originally did not include PDBBind due to our lack of experience with protein-ligand binding affinity prediction. However, as per the reviewer's request, we have fine-tuned our JMP-L backbone on the PDBBind core subset of MoleculeNet (which is from the core set of PDBBind-v2013 and contains binding affinity data for 154 protein-ligand complexes). We report our results in the table below, compared to the relevant results from the original MoleculeNet paper and the most recent results we could find (from https://www.ncbi.nlm.nih.gov/pmc/articles/PMC10301867/). As can be seen, our model achieves SOTA performance on this dataset as well, further demonstrating the generalizability of our model.
> > >
> > > | **Method**                 | **Binding Affinity RMSE** |
> > > | -------------------------- | ------------------------- |
> > > | JMP-L (Ours)               | **1.36**                  |
> > > | DeepBindGCN_RG_x           | 1.49                      |
> > > | SE-OnionNet                | 1.69                      |
> > > | DeepBindRG                 | 1.81                      |
> > > | GraphBAR (dataset 4, best) | 1.63                      |
> > > | BAPA                       | 1.45                      |
> > >
> > >
> > > We thank the reviewer for the suggestion. We are also exploring the v2016 version of PDBBind, but the number of atoms in these systems are too large for our model to handle (up to 25,000 atoms per system). We are currently exploring potential methods (e.g., gradient checkpointing) to facilitate the training of our model on such large systems, and -- if successful -- we will include these results in the final version of the paper.

---

> > > > ### Author Response · Authors · 2023-11-17
> > > > **Response to Reviewer XFKt: 4/5**
> > > >
> > > > > More supervised SOTA should be compared. E.g., for the materials domain, there are two more recent papers [2, 3] that can be reported against the proposed method.
> > > >
> > > > For MatBench, we compare our results to all recent competitive benchmarks on MatBench leaderboard (see https://matbench.materialsproject.org/Leaderboards%20Per-Task/matbench_v0.1_matbench_dielectric/). The papers referenced appear to be trained on MP 2018.6.1 (69,239 materials), which is different from the dataset MP Energy of Formation (132,752 materials) and MP Band Gap (106,113 materials). Link to Matbench dataset can found here (https://matbench.materialsproject.org/Benchmark%20Info/matbench_v0.1/). We have included the numbers below for the sake of comparison but, given the difference in training data, we believe that it would be unfair to include these results in the paper.
> > > >
> > > > | Materials (Units)      | Matformer | PotNet | JMP-S (fold0 / mean) | JMP-L (fold0 / mean) |
> > > > | ---------------------- | --------- | ------ | -------------------- | -------------------- |
> > > > | MP Gap ($eV$)          | 0.211     | 0.204  | 0.119 / 0.121        | 0.089 / **0.091**    |
> > > > | MP E Form ($meV/atom$) | 21.0      | 18.8   | 13.6 / 13.3          | 10.3 / **10.1**      |
> > > >
> > > > > Can you provide more details about the scaling strategy of the model architecture? The message-passing paradigm suffers from over-smoothing a lot and it is notorious of hard to make the network deep. I would like to understand more about this and how this method overcome the issues.
> > > >
> > > > As mentioned in our first comment, due to the high computational cost of our experiments, we made the decision to use GemNet-OC as our backbone architecture due to its favorable properties and diverse application across chemical domains. This model has been shown to scale with increasing number of message passing layers and parameters (see [Gasteiger et al., 2022](https://arxiv.org/abs/2106.08903) and [Gasteiger et al., 2022 (b)](https://arxiv.org/abs/2204.02782)). The original work scales the model up to ~250M parameters (which is the same as our GN-OC-L and JMP-L models). The large models (GN-OC-L and JMP-L) have 6 message passing layers and we did not study the impact of adding more, it is possible that at some point over-smoothing would be an issue, but this is not something that we observed, and, as a matter of fact, we saw a consistent uplift in fine-tuning performance when increasing model parameters (see figure 2 b). We leave scaling model size further for future work.
> > > >
> > > > > Since for larger foundation models, we would like to include far more parameter. But the molecular graphs often just include a very small number of vocabularies. A better strategy could be adding full attention mechanism to make the parameter space larger. And the model would be a transformer-based architecture or so-called "graph transformer". Did the authors try this in their experiments? How did the two frameworks perform?
> > > >
> > > > As mentioned in the previous comment, we made the decision to use GemNet-OC as our backbone architecture due to its favorable properties and diverse application across chemical domains. We agree that a detailed analysis of different backbone architectures would be interesting, but we believe that this is out of scope for this paper, and we plan to explore this in future work. We also agree that a transformer-based architecture would be interesting to explore, and we plan to explore graph-transformer-based architectures, such as EquiformerV2 [[Liao et al., 2023]](https://arxiv.org/abs/2306.12059), in future work.

---

> > > > > ### Author Response · Authors · 2023-11-17
> > > > > **Response to Reviewer XFKt: 5/5**
> > > > >
> > > > > > I understand the concerns of releasing the code and models before acceptance. However, in terms of reproducibility, it would be better to provide an anonymized repo including some demo models for testing purpose.
> > > > >
> > > > > We have uploaded the pre-training and fine-tuning code to an anonymous repository at the [following url](https://anonymous.4open.science/r/jmp-iclr/README-ICLR.md). Our code builds on top of [the Open Catalyst Project's codebase](https://github.com/Open-Catalyst-Project/ocp) and thus re-uses their trainers, dataset implementations (OC20 + OC22), and utility functions. The new additions for our code are:
> > > > > 1. The unified LMDB dataset implementation for both pre-training and fine-tuning datasets (see `ocpmodels/datasets/mt_lmdb.py`).
> > > > > 2. An updated version of GemNet-OC which separates the backbone and the output heads (see `ocpmodels/models/gemnet_oc_mt/`).
> > > > > 3. A new multi-task trainer for pre-training (see `ocpmodels/trainers/mt/`).
> > > > > 4. A new trainer for fine-tuning which provides implementations of our linear warmup + cosine + ReduceLROnPlateau LR scheduling (see `ocpmodels/trainers/ft/`).
> > > > >
> > > > > To use this code, please refer to the `README-ICLR.md` for instructions on setting up the environment, downloading the datasets and pre-trained checkpoints, and running pre-training and/or fine-tuning.

---

### Author Response · Authors · 2023-11-17
**General Response**

We would like to thank all the reviewers for the time and effort they put into giving us feedback! We were glad that all of the reviewers had positive things to say about our work, e.g. “innovative approach” from reviewer `R7em` and “exhaustive and meticulous” from reviewer `avot`, and that they found the paper well written and easy to follow. Moreover, we are happy that all reviewers seemed to be in agreement that the direction of the paper --- pre-training a large model on diverse chemical domains --- is a fruitful and impactful one.

A major critique of our work was about novelty. We agree that the idea of building foundational models for chemistry is obvious and analogous to the progress in other fields such as computer vision and natural language processing. However, we believe our work provides the first convincing demonstration that a foundation model can generalize across chemical domains and be performant on a diverse set of downstream tasks. We achieve SOTA results on 34 of 40 tasks, which span the domains of materials, small molecules, and large molecules. This supports our core hypothesis that diverse pre-training enables beneficial transfer and generalization.

Pre-training models for atomic property prediction has garnered significant interest in the past couple years, with a number of papers published at top venues (see [Zaidi et al., 2023](https://openreview.net/forum?id=tYIMtogyee), [Zhou et al., 2023](https://openreview.net/forum?id=6K2RM6wVqKu)). Nevertheless, most efforts have restricted pre-training to a single chemical domain, and downstream evaluation has been limited hindering our ability to understand generalization. Additionally, previous results have rarely been SOTA, leading to minimal adoption of pre-training. This last point is highlighted by the fact that prior to our work, SOTA on 34 of 40 tasks was held by models trained from scratch.

In this paper, we show that a ML force-field trained with an appropriately formulated supervised pre-training objective can vastly outperform all previous pre-training methods. We lay out the components required for joint multi-domain pre-training and show that the collective impact of these components leads to the success of our approach. The combination of our pre-training approach, the impressive empirical results, and the diverse benchmarking clearly demonstrates the paper "contributes new knowledge and sufficient value to the community" ([2024 ICLR reviewer guidelines](https://iclr.cc/Conferences/2024/ReviewerGuide#:~:text=All%20reviewers%20are%20required%20to,the%20ICLR%20Code%20of%20Ethics.), section 3: Answer four key questions for yourself, to make a recommendation to Accept or Reject).

One additional note, since the time of ICLR submission we found a small bug in our test evaluation code where some runs used an earlier checkpoint instead of the best checkpoint. We have re-run all test evaluations and updated all tables and figures accordingly. We have also updated the introduction to better highlight the challenges of pre-training in the chemical domain and the novelty of our work.

---

### Meta-Review · Area_Chair_ikmL · 2023-12-10

**Metareview:**

This paper introduces an approach for pretraining across multiple datasets for materials property prediction. The authors show that the resulting foundation model outperforms SoTA on a wide variety of tested downstream tasks. While several of the reviewers are concerned that the "novelty" of the ML methods is limited, the ICLR reviewing guidelines state that "Submissions bring value to the ICLR community when they convincingly demonstrate new, relevant, impactful knowledge (incl., empirical, theoretical, for practitioners, etc)." Given that this paper introduces a foundation model for materials property prediction and shows SoTA results across a wide range of datasets widely used within the ICLR community, I believe that this contribution is eminently worth of inclusion at ICLR.

**Justification For Why Not Higher Score:**

Not really applicable, since I've already recommended an increase to this paper's evaluation.

**Justification For Why Not Lower Score:**

See "Additional Comments On Reviewer Discussion."

---

### Decision · Program_Chairs · 2024-01-16

Accept (poster)